# CRISPR screen decodes SWI/SNF chromatin remodeling complex assembly

Hanna Schwaemmle[1,2], Hadrien Soldati[1], Nikolaos M. R. Lykoskoufis [1], Mylène Docquier[2], Alexandre Hainard[3] & Simon M. G. Braun [1,2] ✉

The SWI/SNF (or BAF) complex is an essential chromatin remodeler, which is frequently mutated in cancer and neurodevelopmental disorders. These are often heterozygous loss-of-function mutations, indicating a dosage-sensitive role for SWI/SNF subunits. However, the molecular mechanisms regulating SWI/SNF subunit dosage to ensure complex assembly remain largely unexplored. We performed a CRISPR KO screen, using epigenome editing in mouse embryonic stem cells, and identified *Mlf2* and *Rbm15* as regulators of SWI/SNF complex activity. First, we show that MLF2, a poorly characterized chaperone protein, promotes SWI/SNF assembly and binding to chromatin. Rapid degradation of MLF2 reduces chromatin accessibility at sites that depend on high levels of SWI/SNF binding to maintain open chromatin. Next, we find that RBM15, part of the m[6]A writer complex, controls m[6]A modifications on specific SWI/SNF mRNAs to regulate subunit protein levels. Misregulation of m[6]A methylation causes overexpression of core SWI/SNF subunits leading to the assembly of incomplete complexes lacking the catalytic ATPase/ARP subunits. These data indicate that targeting modulators of SWI/SNF complex assembly may offer a potent therapeutic strategy for diseases associated with impaired chromatin remodeling.

Accurate regulation of gene expression is essential for establishing and maintaining cell identity during embryonic development. Cell-type specific gene expression profiles are established by transcription factors (TFs) via binding to unique DNA motifs. However, TF activity is dependent on non-genetic mechanisms, notably chromatin regulation which impacts DNA accessibility and compaction[1]. Dynamic chromatin states are established by a family of over 400 proteins called chromatin regulators (CRs) that control accessibility and compaction via histone modifications, DNA methylation and nucleosome remodeling[2]. By regulating chromatin accessibility at specific genes, CRs control TF binding to promote differentiation towards diverse cell types during development[3]. The SWI/SNF complex (Switch/Sucrose Non-Fermenting; also known as BAF complex), an ATP-dependent nucleosome remodeling complex, is a major regulator of chromatin accessibility in cells. By evicting and repositioning nucleosomes at enhancers and

promoters, SWI/SNF complexes promote the activation and repression of unique gene expression programs during development[4–6].

Through different combinations of 13 subunits encoded by 29 genes, SWI/SNF complex composition varies depending on cell-type and developmental state. All 29 SWI/SNF genes have been found mutated in children with neurodevelopmental disorders (NDDs)[7]. Furthermore, SWI/SNF complexes are mutated in over 20% of human cancers, making these complexes second only to *TP53* as the most mutated molecular entities in cancer[8]. Pathogenic SWI/SNF mutations are often heterozygous loss-of-function alleles or amplifications, indicating a dosage-sensitive role for SWI/SNF subunits in chromatin regulation. For example, mutations in *ARID1B*, the most frequently mutated gene in NDDs, are predominantly heterozygous frame-shift or missense mutations. These mutations lead to protein destabilization and decreased total ARID1B levels, impairing SWI/SNF activity despite

[1]Department of Genetic Medicine and Development, Faculty of Medicine, University of Geneva, Geneva, Switzerland. [2]Institute of Genetics and Genomics in Geneva (iGE3), University of Geneva, Geneva, Switzerland. [3]Proteomics Core Facility, Faculty of Medicine, University of Geneva, Geneva, Switzerland. ✉e-mail: simon.braun@unige.ch

the presence of an intact WT allele[9]. In addition to *ARID1A/B*, several other SWI/SNF subunits, including *SMARCA2/4*, *SMARCC1/2*, *SMARCE1* and *SMARCB1*, are also haploinsufficient genes, which are defined as genes intolerant to loss of a single allele[10]. Furthermore, overexpression of certain subunits can also impair SWI/SNF activity, as is the case with ACTL6A which is amplified in 25% of squamous cell sarcomas. In these cancers, there is an increase in the incorporation of ACTL6A subunits within SWI/SNF complexes, which promotes an interaction with pro-oncogenic TEAD-YAP transcription factors[11]. Finally, recent studies found that DCAF5, a substrate receptor for E3-ubiquitin ligases, promotes the degradation of incompletely assembled SWI/SNF complexes[12] in SMARCB1-mutant cancers[13]. These studies suggest that maintaining correct stoichiometry of SWI/SNF subunits is important to ensure proper assembly and activity of these multi-subunit complexes. However, the mechanisms that control complex assembly and the relative abundances of SWI/SNF complex subunits within a given cell remain largely unknown.

SWI/SNF chromatin remodeling complexes are highly conserved. For example, yeast and fly SWI/SNF remodelers share similar numbers of subunits, 3D-structures, and biochemical functions with mammalian SWI/SNF complexes[14,15]. Yet, mammalian complexes are distinguished by the greater number of genes encoding diverse subunits[16]. While yeast and flies exhibit only a few different complex assemblies, current estimates suggest that humans can form up to 1400 different SWI/SNF complexes across various tissues[10]. Interestingly, the massive expansion in genome size and regulatory sequences in mammalian genomes coincides with the expansion in the combinatorial assembly of SWI/SNF subunits[17]. With increased subunit diversity, mammalian SWI/SNF complexes have more varied cell-type specific functions during development, enabling unique compositions of the complex to interact with different proteins and arrays of chromatin modifications. To date, several cell-type specific interactors of the SWI/SNF complex have been characterized. However, these account for only a small subset of SWI/SNF's genomic targets throughout development[18–20]. Finally, while much of what we know about SWI/SNF nucleosome remodeling stems from genetic screens in yeast[21,22] and flies[23,24], these pioneering studies lacked the diversity of SWI/SNF complexes to uncover mechanisms which regulate complex assembly and chromatin remodeling activity in mammals.

To bridge this gap, we performed a genome-wide CRISPR screen in mouse embryonic stem cells (mESCs) and uncovered novel regulators of mammalian SWI/SNF complexes during development. First, using epigenome editing tools, we designed a reporter mESC line recapitulating SWI/SNF-dependent gene expression. We then performed a genome-wide forward-genetics knockout screen and identified *Mlf2* and *Rbm15* as required for SWI/SNF activity. We show that MLF2 (Myeloid Leukemia Factor 2), a poorly characterized chaperone protein, regulates a subset of SWI/SNF target genes in mESCs. Rapid MLF2 depletion reduces SWI/SNF chromatin binding, impairing chromatin accessibility at developmental enhancers with high levels of SWI/SNF binding. Next, we show that RBM15 (RNA binding motif 15), a subunit of the $N^6$-methyladenosine (m6A) writer complex, controls selective m6A methylation of SWI/SNF mRNAs to ensure proper protein levels and stoichiometry of SWI/SNF subunits. Misregulation of m6A levels causes overexpression of core SWI/SNF subunits leading to the assembly of incomplete SWI/SNF complexes lacking the catalytic ATPase/ARP subunits. In addition, we find that the RNA recognition motif 1 (RRM1) domain of RBM15 contributes to the targeting of m6A-writer complexes to a subset of mRNAs in mESCs. We propose that post-transcriptional m6A-modifications regulate specific SWI/SNF mRNAs to maintain subunit stoichiometry of mammalian SWI/SNF complexes. Thus, here we decode the intricate mechanisms governing SWI/SNF complex assembly and identify MLF2 and RBM15 as new targets for therapies aimed at modulating SWI/SNF chromatin remodeling activity.

## Results

### A live/dead reporter of SWI/SNF activity in mESCs

To screen for novel regulators of mammalian SWI/SNF complex activity, we designed a reporter of SWI/SNF-dependent gene expression in mouse embryonic stem cells. Previous studies using the FIRE-Cas9 epigenome editing system showed that SWI/SNF-recruitment to the bivalent *Nkx2.9* gene induces transcriptional activation in mESCs[25,26]. Thus, we engineered the *Nkx2.9* locus in mESCs to act as a live/dead reporter of SWI/SNF-dependent gene activation. First, we knocked-in binding motifs for the DNA binding domain of ZFHD1 in the *Nkx2.9* promoter and a Diphtheria Toxin reporter gene (DT-A) in the *Nkx2.9* coding sequence (*ZF-DT-Nkx2.9* mESCs). Next, we stably overexpressed SS18-V5-PYL1 and ZFHD1-ABI1 fusion proteins to inducibly recruit the SWI/SNF complex via its SS18 subunit to the *Nkx2.9* promoter bound by ZFHD1 (Fig. 1a). PYL1 and ABI1 are chemical induced proximity (CIP) tags that dimerize upon abscisic acid (ABA) treatment. Therefore, after ABA treatment we can induce recruitment of SWI/SNF complexes to the *Nkx2.9* promoter leading to expression of DT-A. Previous SWI/SNF recruitment studies have used FRB/FKBP CIP tags that dimerize after Rapamycin treatment[27]. As Rapamycin targets the mTOR pathway and impacts cell proliferation, we switched to ABA, a plant hormone with far fewer off-target effects in mammalian cells[28,29]. To confirm SS18-V5-PYL1 expression led to recruitment of entire SWI/SNF complexes, we performed V5 immuno-precipitation mass-spectrometry (IP-MS) analysis in mESCs and revealed proper incorporation of the fusion protein into SWI/SNF complexes containing all subunits (Supplementary Fig. 1a, b; Supplementary Data 5). Indeed, we pulled down cBAF and ncBAF sub-complexes which contain SS18, but not the pBAF sub-complex. We also performed V5 and SMARCC1 chromatin immunoprecipitation (ChIP) experiments and detected high levels of SWI/SNF complex recruitment to the target locus (Fig. 1b). Furthermore, we detected increased chromatin accessibility by ATAC-qPCR following SWI/SNF recruitment to the *Nkx2.9* locus (Supplementary Fig. 1c), validating the cell line as a reporter of SWI/SNF chromatin remodeling activity. Finally, we selected DT-A as a reporter gene for SWI/SNF activity, as its activation leads to Diphtheria Toxin (DT) mediated cell death via inhibition of translation. We show that over 95% of *ZF-DT-Nkx2.9* mESCs die following 96 h of ABA treatment to induce SWI/SNF activity (Fig. 1c). The robust live/dead reporter of SWI/SNF activity was ideal for our subsequent loss-of-function genetic screen as it offered key advantages over FACS-based approaches that require sorting of millions of cells, often with low signal-to-noise ratios.

### Genome-wide CRISPR screen reveals SWI/SNF activity regulators

We used the GeCKOv2 CRISPR lentivirus library[30] to perform a genome-wide knockout screen in the reporter *ZF-DT-Nkx2.9* mESCs. This lentivirus library is split in two, each library contains 3 sgRNAs targeting all mouse genes as well as control sgRNAs. We infected *ZF-DT-Nkx2.9* mESCs with each lentivirus library at an MOI of <0.4x, to obtain single knock-out mESCs with a ~1000x coverage. We treated the lentivirus infected cells with Puromycin for several days to select for knock-outs. The pooled knock-out *ZF-DT-Nkx2.9* mESCs were then treated with ABA for 48 h to induce SWI/SNF activity and cell death. We collected the cells that survived the ABA treatment, as well as control cells not treated with ABA and performed next generation sequencing to identify the enriched sgRNAs (Fig. 1d). In control conditions we detected over 97% of all sgRNAs for each library, detecting 100–1000 reads for the vast majority of sgRNAs (Supplementary Fig. 1d), which confirmed the coverage and diverse representation of the >130'000 different sgRNAs in the screen. We used the MAGeCK software[31] to identify sgRNAs that were enriched in the ABA treated conditions. We found 32 genes that were enriched in both libraries with multiple sgRNAs, using a stringent cut-off enrichment score of 1.00E-04 (Fig. 1e, Supplementary Fig. 1e and Supplementary Data 1). Of note, we

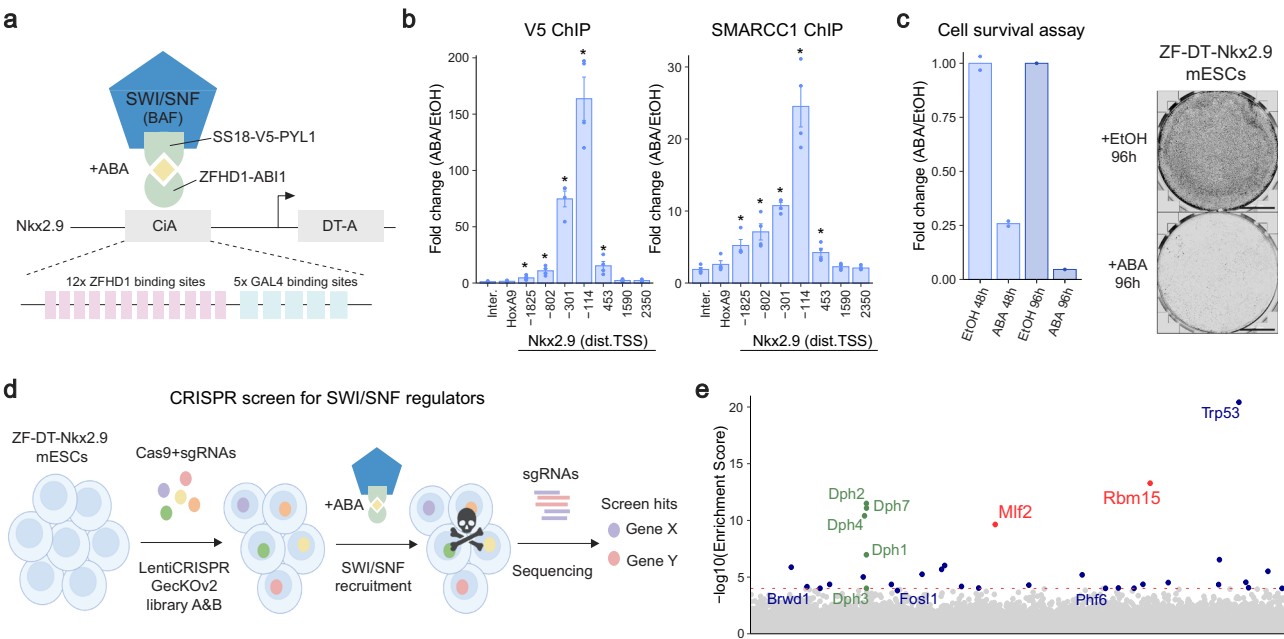

**Fig. 1 | Genome-wide CRISPR KO screen reveals novel regulators of SWI/SNF activity. a** Schematic of the epigenome editing strategy used to engineer the *ZF-DT-Nkx2.9* mESC line as a live/dead reporter of SWI/SNF activity. After abscisic acid (ABA) treatment SS18-V5-PYL1 dimerizes with ZFHD1-ABI1 fusion proteins that are bound to ZF motifs knocked into the *Nkx2.9* promoter, downstream of the DT-A reporter gene. Created in BioRender https://BioRender.com/ztebfa0. **b** ChIP analysis of SWI/SNF complex recruitment to the *Nkx2.9* promoter after 24 h of ABA treatment. V5-ChIP shows SS18-V5 binding to the target locus. SMARCC1-ChIP shows recruitment of assembled SWI/SNF complexes. qPCR analysis using primer pairs tilling the *Nkx2.9* locus with the distance from transcription start site (TSS) indicated, and control primers for an intergenic region and *HoxA9*. Depicted are fold change values (mean ± SEM) for $n = 4$. Statistical significance calculated using two-sided *t*-test (significant *p*-values form left to right: V5-ChIP $p = 0.05$, $p = 0.02$, $p = 0.002$, $p = 0.004$, $p = 0.04$; SMARCC1 ChIP $p = 0.02$, $p = 0.016$, $p = 1.9e\text{-}5$,

$p = 0.004$, $p = 0.02$). **c** Timecourse of cell survival after ABA treatment in *ZF-DT-Nkx2.9* mESCs ($n = 2$ for 48 h, $n = 1$ for 96 h). Giemsa staining for viable cells after ABA or EtOH control treatment for 96 h; scale = 1 mm. **d** Schematic of CRISPR knock-out screening strategy using the GeCKOv2 libraries. *ZF-DT-Nkx2.9* mESCs were infected with library A or B at an MOI of <0.4x followed by puromycin selection. SWI/SNF activity was induced by ABA treatment for 48 h. Surviving cells were collected, genomic DNA was isolated and sgRNA sequences were identified by PCR and sequencing. Created in BioRender https://BioRender.com/owp27nz. **e** Plot displays gene enrichment score from MAGeCK analysis of sgRNA counts between ABA and EtOH treated cells. Red dotted line corresponds to a stringent enrichment score cut-off of 1.00E-04. Genes required for the biosynthetic pathway of diphthamide are labeled in green. Gene names in blue are known regulators of the SWI/SNF complex and gene names in red are novel regulators. Source data are provided as a Source Data file.

identified *Dph1*, *Dph2*, *Dph3*, *Dph4* and *Dph7* amongst the top hits in the screen. These genes are required in the biosynthetic pathway of diphthamide, the molecular target of Diphtheria Toxin[32]. By enriching for genes in this pathway, we validated our live/dead screening system. In addition, we identified previously known interactors of SWI/SNF such as *Brwd1*[33], *Fosl1*[34], *Phf6*[35,36] and *Trp53*[37]. Furthermore, previous studies have performed genetic screens aimed at identifying genes required for DT-mediated cell death[38,39]. We therefore cross-referenced our hits with these screens to only select for further analysis genes relevant to SWI/SNF activity rather than DT-mediated cell death. We selected 9 enriched genes (*Dph2*, *Mlf2*, *Rbm15*, *Hopx*, *Tfap2c*, *Gna13*, *Setd1b*, *Kdm6a* and *Tet1*) for further validation as candidate regulators of SWI/SNF activity during mammalian development.

## Validation of selected hits from the genetic screen

To validate the 9 selected hits of the genetic screen we used CRISPR/Cas9 to generate individual gene knock-outs in the *ZF-DT-Nkx2.9* mESC reporter line (Fig. 2a, b). For each gene we selected two sgRNAs targeting the same exon to generate indels. We confirmed homozygous gene editing in all clonal cell lines by PCR and sanger sequencing, as well as loss of protein expression by western blot for 4/9 hits with working antibodies (Fig. 2e and Supplementary Fig. 2a–c). We then treated each cell line with ABA for 48 h and quantified the number of surviving cells. As a control we used the *ZF-DT-Nkx2.9* mESC line transfected with non-targeting sgRNAs (NT). We found that 5 out of 9 hits (*Dph2*, *Mlf2*, *Rbm15*, *Setd1b*, and *Tet1*) showed significant increases in cell survival compared to controls, suggesting that the genes are required for SWI/SNF-

dependent gene activation (Fig. 2c). Previous studies have shown that SWI/SNF-dependent bivalent gene activation is characterized by rapid losses of H3K27me3 repressive histone marks coupled with rapid gains in H3K4me3 activating histone marks, that cannot only be explained by histone H3 turnover[25–27]. These epigenetic changes occur within minutes of SWI/SNF recruitment before transcriptional changes can be detected, hinting at an active process that requires demethylase and methyltransferase activity. Interestingly, the H3K27me3 demethylase *Kdm6a*[40], the H3K4me3 methyltransferase *Setd1b*[41] and the DNA demethylase *Tet1*[42] were enriched in the knockout screen for SWI/SNF-dependent regulators of *Nkx2.9* activation. In validation experiments, *Setd1b* KO and *Tet1* KO mESCs partially rescued the cell death phenotype, whereas *Kdm6a* KO mESCs showed a modest, non-significant increase in cell survival (Fig. 2c). These results suggest that SETD1B and TET1 function in concert with SWI/SNF complexes to promote an active epigenetic landscape and drive *Nkx2.9* gene expression. The most robust rescue of cell death was observed for two of the top hits of the screen, *Mlf2* and *Rbm15* (Fig. 2c). Indeed, the majority of *Mlf2* KO and *Rbm15* KO *ZF-DT-Nkx2.9* mESCs survived ABA treatment compared to NT controls (Fig. 2d, f). Thus, we next sought to investigate how these genes regulate SWI/SNF complex activity in mESCs.

## MLF2 regulates a subset of SWI/SNF target genes in mESCs

MLF2 (Myeloid Leukemia Factor 2) is a poorly characterized ubiquitously expressed protein that is mutated in human cancers[43–45]. Previous work investigating MLF (the fly homolog of MLF1 and MLF2) showed that in flies, MLF acts in a chaperone complex with HSP70 to

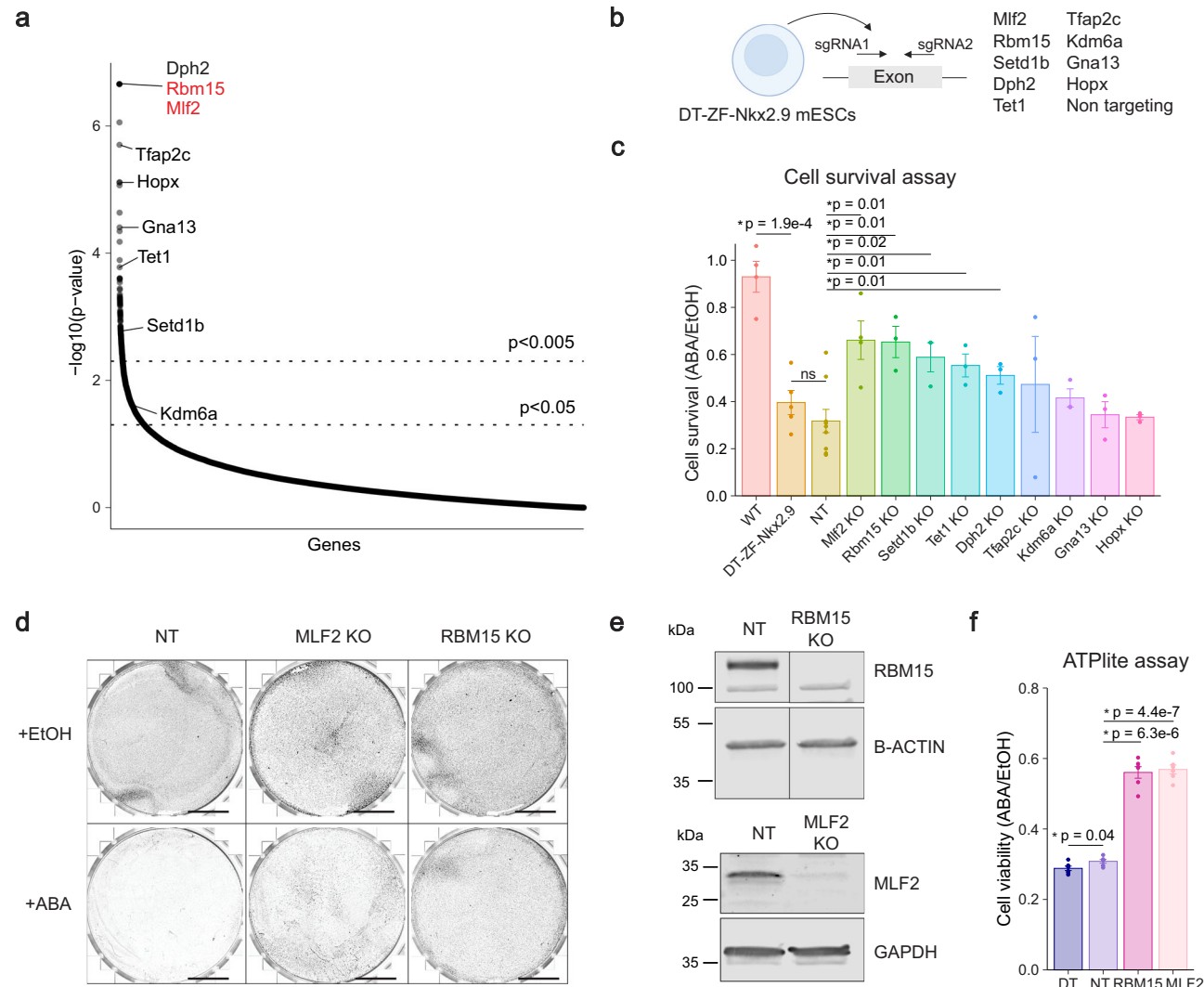

**Fig. 2 | Validation of top hits from genetic screen in mESCs. a** Distribution of hits identified in CRISPR screen ranked by *p*-value (permutation test using MAGeCK software, see Methods). **b** Schematic of CRISPR KO mESC strategy. Two guides targeting an exon were used to induce a frameshift indel mutation of the target gene. To generate 9 clonal KO cell lines, individual mESC colonies were picked and amplified, then target gene disruption was confirmed by PCR and sanger sequencing. A control cell line was generated using non-targeting sgRNAs (NT). Created in BioRender https://BioRender.com/g6p0ul6. **c** Graph shows cell survival for each KO mESC line as fold change of 48 h ABA treatment compared to EtOH treated control cells (mean ± SEM). Statistical significance calculated using two-sided *t*-test, replicates *n* = min(3). **d** Giemsa staining for viable cells after 96 h of ABA / EtOH treatment. Mlf2 KO and Rbm15 KO mESCs show increased survival compared to NT controls; scale = 1 mm; *n* = 1. **e** Western blot analysis of RBM15 and MLF2 levels in NT and KO mESC lines. B-ACTIN and GAPDH as loading controls (*n* = 1). **f** Plot shows ATPlite cell viability assay results as fold change following 48 h ABA treatment in ZF-DT-Nkx2-9 mESCs, NT control, Rbm15 KO and Mlf2 KO ZF-DT-Nkx2.9 mESCs over 48 h EtOH treatment (mean ± SEM). Statistical significance calculated using two-sided *t*-test, replicates *n* = 6. Source data are provided as a Source Data file.

regulate the stability of transcription factors and impact gene expression[46]. Interestingly, *Hspa4* (Heat shock protein family A HSP70 member 4) was also amongst the top enriched genes in our CRISPR screen (Fig. 3a), suggesting a similar chaperone mechanism may regulate chromatin regulator stability in mESCs. To study the direct function of MLF2 in mESCs, we took advantage of the dTAG degron system[47] that allows for rapid degradation of target proteins via small molecule recruitment of E3-ubiquitin ligases (Fig. 3b). Using CRISPR/Cas9 genome editing, we knocked-in the dTAG degron tag (FKBPv-V5) at the C-terminus of MLF2 to generate MLF2-dTAG mESCs. In addition, to accurately compare acute MLF2 degradation to acute SWI/SNF degradation, we generated SMARCA4-dTAG mESCs. We chose the C-terminus of the SMARCA4 subunit as it is the catalytic ATPase that is required for SWI/SNF chromatin remodeling activity. Addition of the small molecule dTAG-13 to MLF2-dTAG or SMARCA4-dTAG cells led to rapid proteasomal degradation of tagged proteins. Within 3 h of dTAG-13 treatment, MLF2 or SMARCA4 were no longer detected by western

blot (Fig. 3c). The insertion of dTAG degron tags slightly decreased MLF2 but not SMARCA4 levels compared to WT mESCs (Supplementary Fig. 3a). However, confocal imaging confirmed the correct subcellular location of the tagged proteins. SMARCA4 was abundant in the nucleus, whereas MLF2 expression was detected in both the cytoplasm and nucleus (Supplementary Fig. 3b). As shown previously, prolonged SMARCA4 loss was lethal for mESCs after 3 days[5]. In contrast, prolonged dTAG treatment in MLF2-dTAG cells caused only a modest reduction in cell proliferation after 6 days (Supplementary Fig. 3c). To determine if MLF2 regulates the same target genes as SWI/SNF, we conducted RNA-seq experiments after 3 h, 8 h and 24 h of dTAG-13 treatment in both cell lines (Fig. 3d, e and Supplementary Data 2). As expected, SMARCA4 degradation led to major changes in gene expression which increased over time. As a control, we performed RNA-seq experiments in WT mESCs treated with dTAG-13 for 24 h and detected only seven differentially expressed genes (FDR < 0.05 and fold change > 1.5), demonstrating that dTAG-13 alone has a negligeable

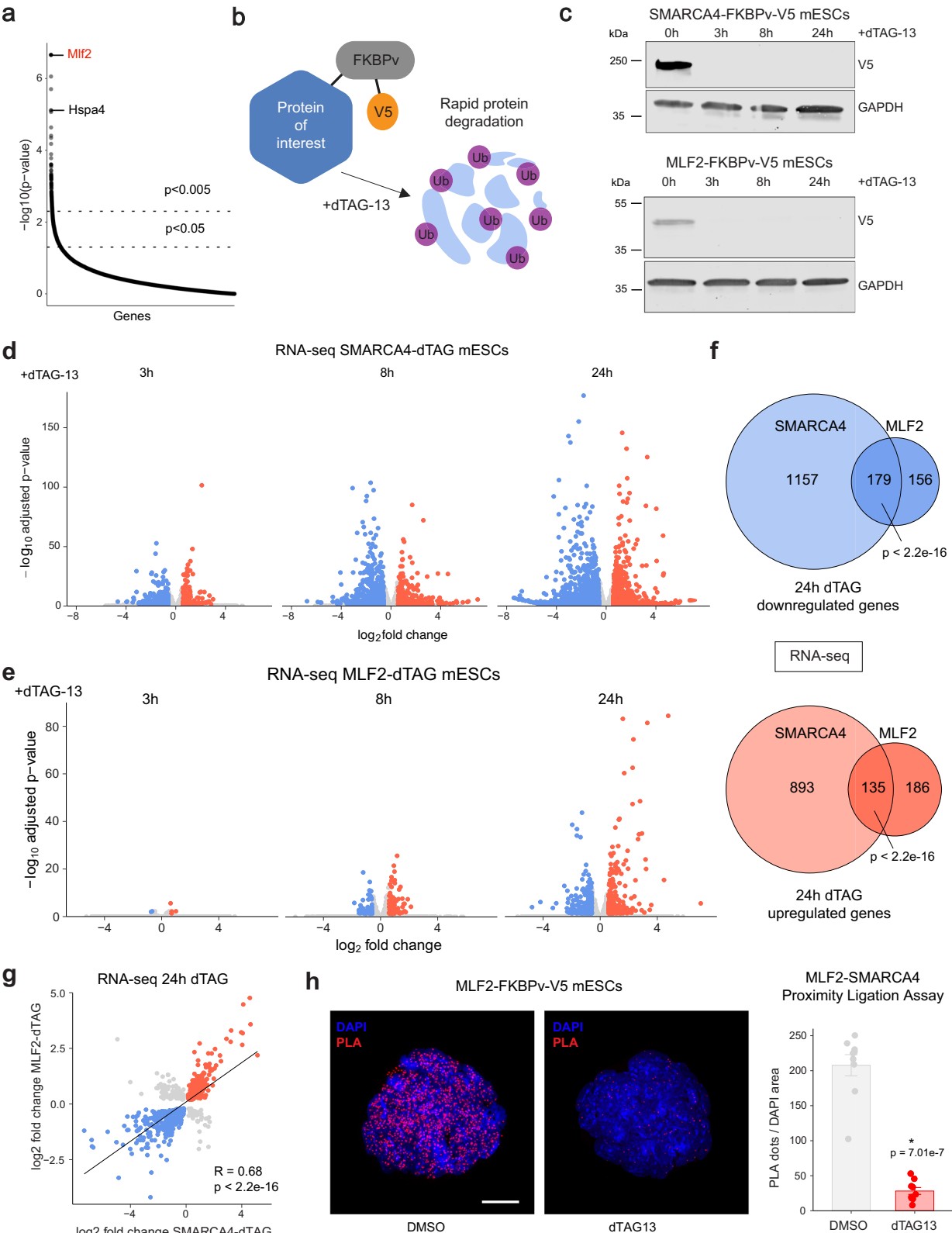

impact on mESC gene expression (Supplementary Fig. 3d). Interestingly, we detected time-dependent changes in gene expression following acute MLF2 depletion, suggesting a role for MLF2 in transcriptional regulation. Next, we investigated whether transcriptional changes gated by both proteins overlap. While the number of differentially regulated genes was higher in SMARCA4-dTAG cells (2'382 genes after 24 h) compared to the MLF2-dTAG cells (656 genes

after 24 h), we detected a strong overlap between the two datasets (Fig. 3f, g and Supplementary Fig. 3f, g). Almost 50% of MLF2 target genes were also misregulated in SMARCA4-dTAG mESCs, suggesting that MLF2 regulates a subset of SWI/SNF target genes. Furthermore, Gene Ontology (GO-term) analysis revealed that MLF2 and SMARCA4 target genes were enriched in the same pathways (Supplementary Fig. 3e). For example, the overlapping downregulated genes are

**Fig. 3 | MLF2 regulates a subset of SWI/SNF target genes in mESCs. a** Plot displays ranked *p*-values (permutation test using MAGeCK software, see Methods) of enriched genes in KO screen. Labeled genes are top hits *Mlf2* and the HSP70 chaperone complex gene *Hspa4*. **b** Schematic of dTAG-degron system. The target protein is endogenously tagged with FKBPv-V5. Upon addition of dTAG-13, the proteasomal machinery is recruited to the FKBPv tag, the target protein is ubiquitylated and rapidly degraded via the proteasome. Created in BioRender https://BioRender.com/hatsa1b **c**. Western blot of SMARCA4 and MLF2 dTAG mESCs showing time course of dTAG-13 treatment vs DMSO control. Both SMARCA4 and MLF2 were detected using a V5 antibody (*n* = 1). **d** Volcano plots showing differentially expressed genes in SMARCA4-dTAG mESCs after 3 h, 8 h, and 24 h of dTAG-13 or DMSO treatment. Significantly downregulated genes are in blue and upregulated in red (sig. = padj. <0.05 & |FC | > 1.5; *n* = 3). Adjusted *p*-values were calculated using the Benjamini-Hochberg correction using DESeq2 software, see Methods. **e** Volcano plots showing differentially

expressed genes in MLF2-dTAG mESCs after 3 h, 8 h, and 24 h of dTAG-13 or DMSO treatment. Significantly downregulated genes are in blue and upregulated in red (sig. = padj. <0.05 & |FC | > 1.5; *n* = 3). Adjusted *p*-values were calculated using the Benjamini-Hochberg correction using DESeq2 software, see Methods. **f** Venn diagrams showing the overlapping genes after 24 h dTAG-13 treatment in SMARCA4 and MLF2 dTAG mESCs. Statistical significance calculated using a two-sided Fisher's exact test. **g** Dot plot of significantly (*p* < 0.05) changed genes, overlapping in the 24 h dTAG-13 datasets of SMARCA4 and MLF2 dTAG mESCs. The linear regression line was fitted using Pearson correlation, *p*-value was calculated using a two-sided *t*-test. **h** Confocal images of proximity ligation assay used to detect MLF2-V5 and SMARCA4 interactions in MLF2-dTAG mESCs treated with DMSO or dTAG-13 for 24 h. Nuclei stained with DAPI. Graph shows number of PLA dots detected in both conditions, as mean ± SEM. Statistical significance calculated using a two-sided *t*-test, replicates *n* = 9, scale bar = 15 μm. Source data are provided as a Source Data file.

involved in embryonic development and BMP signaling pathways, two well characterized SWI/SNF-dependent pathways[48], indicating a role for MLF2 in mESC pluripotency and differentiation. Next, to measure protein-protein interactions in situ we performed Proximity Ligation Assays (PLA) using antibodies against V5-MLF2 and SMARCA4 (Fig. 3h). Using confocal microscopy, we detected strong interactions between MLF2 and SMARCA4 in the nucleus of MLF2-dTAG cells treated with DMSO. As a control, we depleted MLF2 via dTAG-13 treatment and observed a major reduction in the PLA signal (Fig. 3h). Finally, we measured SWI/SNF subunit protein levels in MLF2-dTAG mESCs treated with dTAG-13 for 8 h. For the subunits tested (SMARCA4, SMARCC1, SMARCD1, SMARCE1 and ARID1A) we did not detect major differences in MLF2-depleted cells compared to controls (Supplementary Fig. 3h). These data suggest that MLF2 interacts with SWI/SNF complexes in the nucleus, where it acts as a chaperone to promote complex assembly and stability rather than the abundance of individual subunits.

## MLF2 regulates SWI/SNF chromatin remodeling activity in mESCs

To understand if MLF2 has a direct impact on SWI/SNF chromatin remodeling activity or if it regulates transcription downstream of SWI/SNF function, we performed ATAC-seq experiments in MLF2-dTAG and SMARCA4-dTAG mESCs treated with dTAG-13 for 8 h. As expected, acute SMARCA4 degradation led to dynamic losses in chromatin accessibility at thousands of sites across the genome, corroborating SWI/SNF's major role in chromatin regulation (Fig. 4a and Supplementary Data 3). Interestingly, acute MLF2 degradation also had a direct impact on chromatin accessibility at thousands of loci in mESCs (Fig. 4a and Supplementary Fig. 4a). We observed a strong overlap in differentially regulated peaks between both datasets, with 50% of the 2'111 downregulated ATAC-seq peaks in MLF2-dTAG cells also downregulated in SMARCA4-dTAG cells (Fig. 4b, d; FDR < 0.05 and fold change > 1.5). Moreover, down-regulated ATAC-seq peaks from both conditions were most predominantly detected at regulatory enhancer sequences enriched for pluripotency factor binding sites POU5F1, SOX2 and NANOG (Fig. 4c and Supplementary Fig. 4b). GO-term analysis of genes associated to downregulated ATAC-seq peaks showed an enrichment for genes involved in neurodevelopmental processes (Supplementary Fig. 4c) in both conditions. These data suggest that MLF2 directly regulates SWI/SNF chromatin remodeling activity. To test if MLF2 is required for SWI/SNF binding to chromatin we performed SMARCA4 CUT&RUN experiments in MLF2-dTAG mESCs treated with dTAG-13 for 8 h. Indeed, MLF2-loss significantly reduced SMARCA4 binding genome-wide (Fig. 4e). To accurately assess these global losses in SMARCA4 binding across the three replicates we used spike-in controls in each sample (Supplementary Fig. 4d). Next, we performed MLF2-V5 ChIP-seq in mESCs using the V5-tag inserted with the dTAG degron system. We did not detect MLF2 bound to chromatin across the genome

(Supplementary Fig. 4e), despite using the same V5 ChIP antibody as previously used to detect high levels of V5-SS18 recruitment to the *Nkx2.9* locus in *ZF-DT-Nkx2.9* mESCs (Fig. 1c). Furthermore, we performed sequential salt extraction assays on cell extracts and detected MLF2 in the cytoplasmic and nuclear fractions but not in any of the chromatin bound fractions, confirming that MLF2 does not bind chromatin (Supplementary Fig. 4f). Finally, we analyzed the SMARCA4 CUT&RUN data together with published SMARCA4 ChIP-seq[49] data from mESCs and found that MLF2-dependent ATAC-seq peaks were located at sites with high levels of SMARCA4 binding. We sorted SMARCA4 peaks into 5 clusters according to peak intensity and found that more than 70% of the downregulated ATAC-seq peaks in MLF2-dTAG cells treated with dTAG-13 were at sites with high levels of SMARCA4 (Fig. 4f, g and Supplementary Fig. 4g). Taken together, our data suggests that the chaperone protein MLF2 regulates binding but not specific targeting of SWI/SNF to chromatin, by modulating upstream processes such as SWI/SNF complex stability and assembly. Thus, loss of MLF2 impacts chromatin accessibility at a subset of developmental enhancers by affecting sites that are most dependent on high levels of SWI/SNF binding.

## RBM15 regulates m6A modifications on specific SWI/SNF mRNAs

Several members of the $N^6$-methyladenosine (m6A) RNA methylation pathway were enriched in our genetic screen including *Rbm15*, *Virma*, *Ythdf1*, *Ythdf2* and *Ythdf3* (Fig. 5a). m6A is a prevalent epigenetic modification placed on mRNAs to regulate the cellular fate of modified transcripts[50,51]. These marks are catalyzed by the m6A-writer complex comprised of seven subunits: the methyltransferases METTL3/14 and the regulatory subunits VIRMA, WTAP, HAKAI, ZC3H13 and RBM15/15B (Fig. 5a). After methylation, m6A reader proteins like the YTHDF1/2/3 and IGF2BP1/2/3 families regulate the stability and translation efficiency of m6A-modified mRNAs[52]. To validate that this pathway is indeed required for SWI/SNF-dependent gene expression, we treated *ZF-DT-Nkx2.9* mESCs with a specific chemical inhibitor of METTL3 (STM2457[53]) and observed a significant increase in cell viability following ABA treatment compared to untreated controls (Fig. 5b). Next, we focused on the role of RBM15 as it was the second highest enriched gene in the screen. RBM15 has three N-terminal RNA Recognition Motifs (RRMs) as well as a Spen orthologue and paralogue C-terminal (SPOC) domain. As the only subunits of the m6A-writer complex with RRMs, it is thought that RBM15 and RBM15B may be required to recognize specific sites on mRNAs[54]. We generated clonal *Rbm15* KO mESCs from WT cells using CRISPR/Cas9 and confirmed loss of RBM15 protein by western blot (Fig. 5c). To determine RBM15 target genes in mESCs, we performed RNA-seq on purified mRNAs from non-targeting control (NT) cells and *Rbm15* knockout (KO) cells (Fig. 5d and Supplementary Data 4). We detected 833 downregulated and 1064 upregulated genes (padj <0.05 & |FC | > 1.5), suggesting that like m6A methylation, RBM15 acts in a dual manner and can either increase or decrease certain mRNA levels. GO-term analysis revealed that RBM15

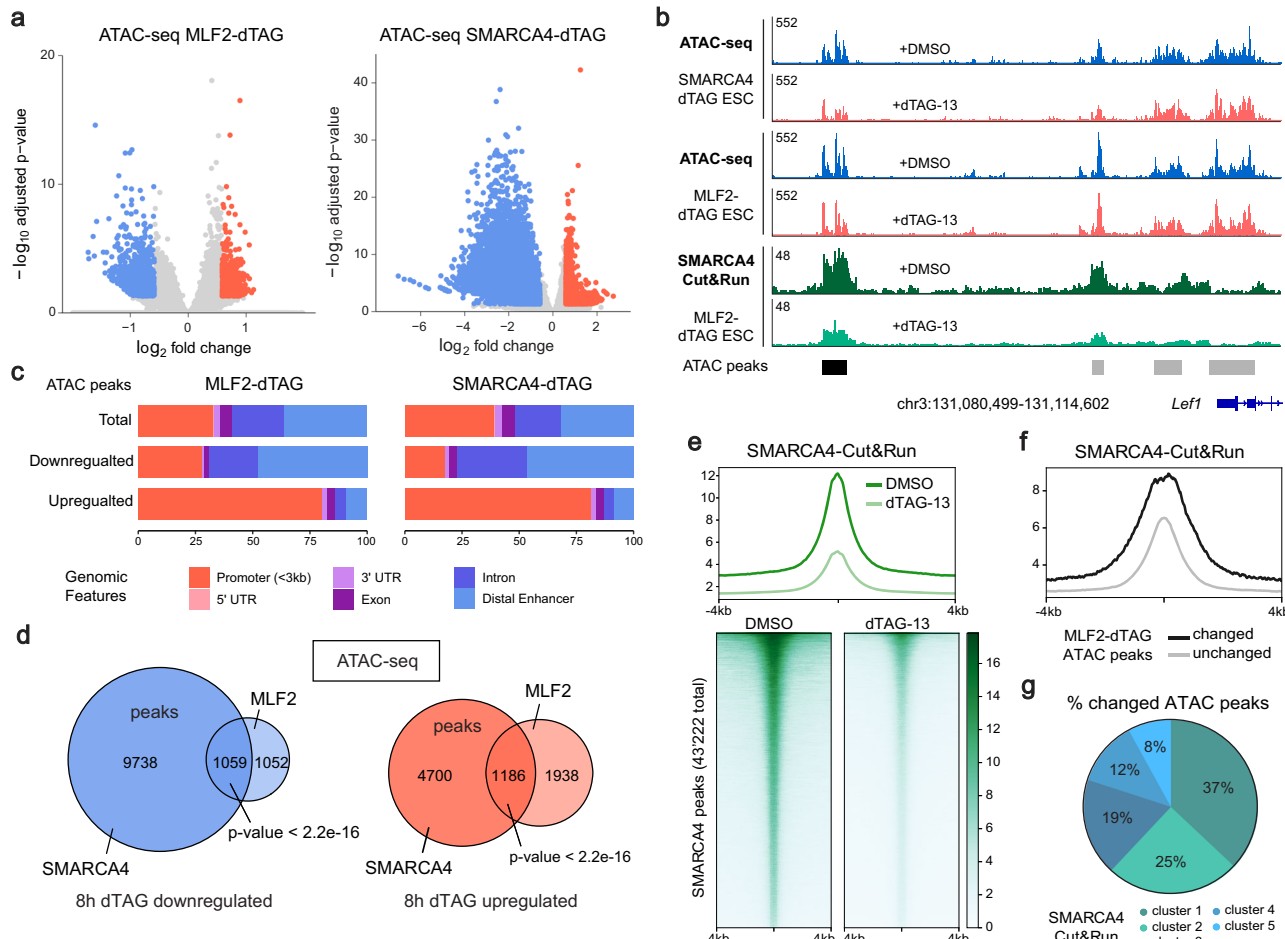

**Fig. 4 | MLF2 regulates SWI/SNF chromatin remodeling activity in mESCs.**
**a** Volcano plots showing differentially accessible peaks in SMARCA4 and MLF2 dTAG mESCs after 8 h dTAG-13 vs DMSO treatment. Significantly downregulated peaks are labeled in blue and upregulated in red (sig. = padj. <0.05 & |FC| > 1.5; $n = 3$). Adjusted $p$-values were calculated using the Benjamini-Hochberg correction using DESeq2 software, see Methods. **b** Genome browser tracks of ATAC-seq peaks at the *Lef1* locus for SMARCA4 and MLF2 dTAG mESCs after 8 h dTAG-13 (red) and DMSO (blue) treatment. SMARCA4 Cut&Run tracks from MLF2 dTAG mESCs show SWI/SNF occupancy at the *Lef1* locus (green) in DMSO conditions and after MLF2 loss in dTAG-13 treated conditions. The significantly downregulated ATAC peaks in both SMARCA4 and MLF2 dTAG mESCs are labeled in black. **c** Genomic features of

ATAC peaks in SMARCA4 and MLF2 dTAG mESCs after 8 h dTAG-13 treatment. **d** Venn diagrams showing the overlapping ATAC peaks at 8 h dTAG-13 treatment for SMARCA4 and MLF2 dTAG mESCs. Statistical significance was calculated using a two-sided Fisher's exact test. **e** SMARCA4 Cut&Run signal in MLF2 dTAG cells with 8 h treatment of DMSO or dTAG-13, centered at the peak center and ranked according to read intensity. **f** SMARCA4 Cut&Run signal at changed and unchanged ATAC peaks in MLF2-dTAG mESCs after 8 h of dTAG-13 treatment. **g** SMARCA4 Cut&Run data was clustered into 5 clusters with equal numbers of peaks, ranging from high occupancy (1) to low occupancy (5). Graph shows the % of changed ATAC-seq peaks in MLF2-dTAG mESCs in each cluster.

target genes were enriched for developmental processes and neural differentiation pathways (Supplementary Fig. 5e), in line with previous reports from KO studies of other m⁶A writer components in mESCs[55,56]. To directly measure the impact of RBM15 loss on m⁶A distribution we performed m⁶A RNA immunoprecipitation sequencing (m⁶A-RIP-seq[55,57]) on purified mRNAs (Fig. 5e and Supplementary Data 4). As expected, we detected a strong enrichment of m⁶A modifications in the 3'UTR and exons at the 3' end of transcripts (Fig. 5f). We found that RBM15 loss caused dynamic changes in m⁶A modifications across thousands of mRNAs, with 1'943 peaks showing decreased m⁶A levels and 2'827 peaks showing increased m⁶A levels (padj <0.05 & |FC| > 1.5). These data suggest that in the absence of the RBM15 subunit, the m⁶A-writer complex can still methylate mRNAs but target selection is misregulated as certain sites gain m⁶A modifications whereas others lose them. To further test the hypothesis that accuracy of the writer complex is controlled by RBM15, we disrupted one of the three RNA recognition motif (RRM) domains in RBM15 by CRISPR/Cas9 in mESCs. Western blot analysis confirmed similar expression of RBM15 ΔRRM1 mutant protein compared to that in NT control cells (Fig. 5c and

Supplementary Fig. 5f). To determine RRM1-dependent target genes in mESCs, we performed RNA-seq and m⁶A-RIP-seq in RBM15 ΔRRM1 mESCs (Supplementary Fig. 5a–d). We detected 265 downregulated and 253 upregulated genes (padj <0.05 & |FC| > 1.5), as well as 232 peaks showing decreased m⁶A levels and 423 peaks showing increased m⁶A levels (padj <0.05 & |FC| > 1.5). We observed a significant correlation in the mRNAs and m⁶A peaks that were altered in RBM15 ΔRRM1 and RBM15 KO cells (Fig. 5g, h). Strikingly, although fewer mRNAs were impacted, there was a remarkable overlap in both the mRNAs (87%) and m⁶A peaks (93%) which were misregulated in ΔRRM1 mutant and full KO cells (Fig. 5i). These results suggest that the RRM1 domain of RBM15 contributes to the targeting of m⁶A-writer complexes to a specific subset of mRNAs in mESCs. A previous study has shown that *RBM15* knockdown in HEK293 human cancer cell lines increases protein levels of the SWI/SNF subunit SMARCC1[58]. Therefore, we sought to determine whether RBM15 targets SWI/SNF subunit mRNAs to regulate SWI/SNF complex activity in mESCs. Indeed, we found that while many SWI/SNF subunit genes have m⁶A modifications, only a specific set of these SWI/SNF mRNAs showed changes in m⁶A distribution in RBM15

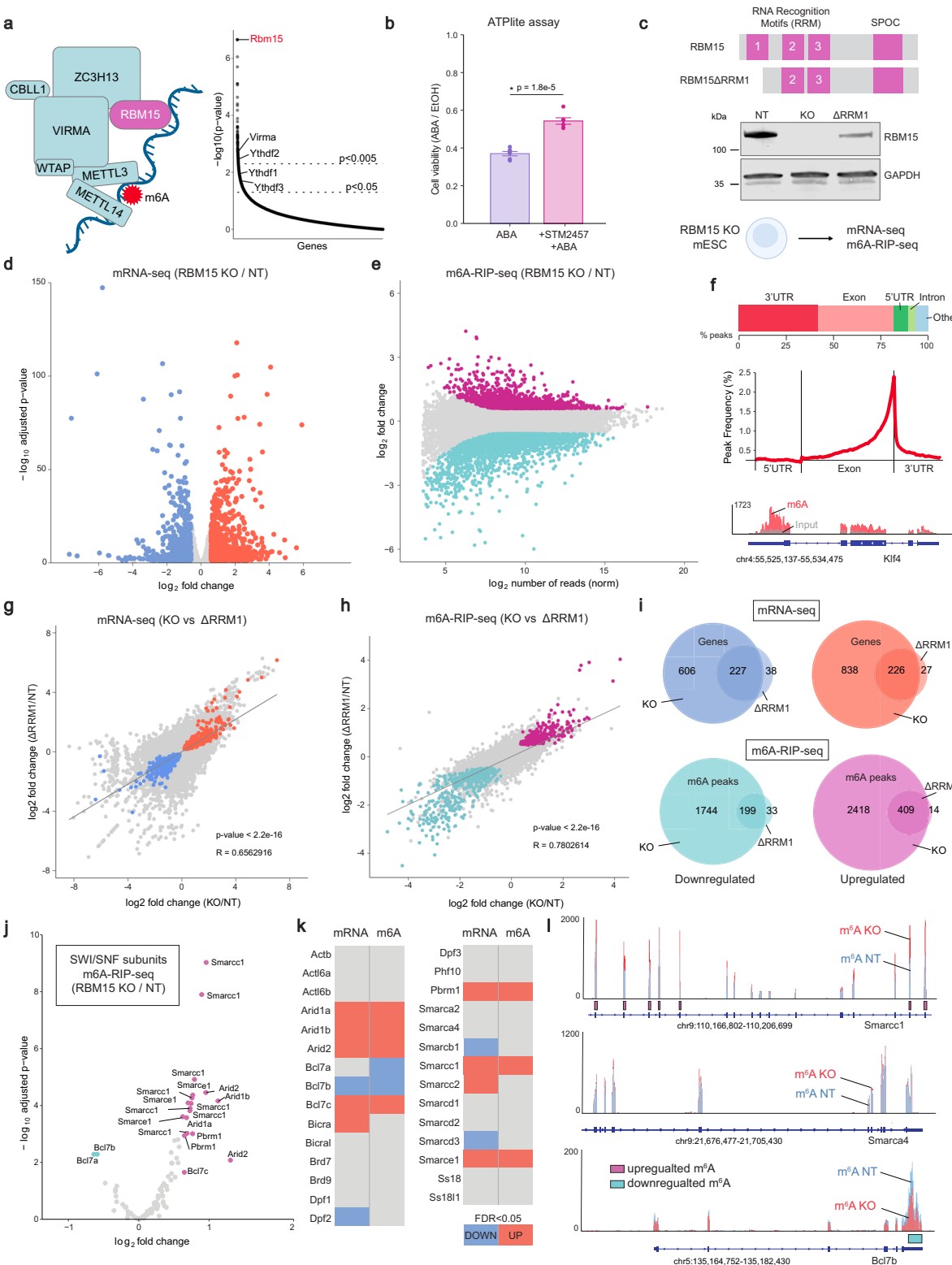

KO cells (Fig. 5j). For example, the SWI/SNF subunits *Smarcc1*, *Arid1a/b* and *Smarce1* displayed significantly increased levels of m6A methylation at specific exons. Whereas the *Bcl7b* subunit had reduced levels of m6A methylation within its 3' UTR and *Smarca4* had no changes in m6A distribution. (Fig. 5k, l). These data suggest that RBM15 regulates SWI/SNF activity via regulation of m6A deposition on mRNAs of specific SWI/SNF subunits.

## RBM15 regulates SWI/SNF complex subunit stoichiometry

Upon mRNA export to the cytoplasm, m6A modifications are bound by reader proteins (ex: YTHDF1/2/3) that regulate mRNA stability and translation rates (Fig. 6a). We hypothesized that misregulation of m6A levels on SWI/SNF mRNAs in RBM15 KO mESCs may therefore alter subunit protein levels, leading to impaired SWI/SNF complex activity. As with many multi-subunit complexes, SWI/SNF complex assembly

**Fig. 5 | RBM15 controls m⁶A methylation on specific SWI/SNF mRNAs.**
**a** Schematic of the m⁶A RNA methylation complex. Plot displays ranked *p*-values (permutation test using MAGeCK software, see Methods) of enriched genes in KO screen. Labeled genes are members of m⁶A RNA methylation pathway enriched in the screen. Created in BioRender https://BioRender.com/255etvx. **b** Plot shows fold change of ATPlite cell viability assay results following 48 h ABA treatment in ZF-DT-Nkx2-9 mESCs cultured with STM2457 m⁶A inhibitor (1 μM) or DMSO over EtOH treated controls for 96 hours, presented as mean ± SEM. Statistical significance calculated using two-sided *t*-test, replicates *n* = 6. **c** Schematic of RBM15 protein domains and western blot analysis of RBM15 levels in NT, KO and ΔRRM1 mESCs. GAPDH as loading control (*n* = 1). Created in BioRender https://BioRender.com/887opbj. **d** Volcano plot showing differentially expressed genes (DEGs) between NT and KO mESCs (*n* = 2, padj <0.05 and |FC| > 1.5). The downregulated genes are in blue and the upregulated genes are in red. Adjusted *p*-values were calculated using the Benjamini-Hochberg correction using DESeq2 software, see Methods. **e** MA plot showing differentially m⁶A modified mRNA sites between NT and KO mESCs (*n* = 2, padj <0.05 and |FC| > 1.5). The downregulated peaks are in cyan and the upregulated peaks are in magenta. Adjusted *p*-values were calculated using the Benjamini-Hochberg correction using DESeq2 software, see Methods. **f** Bar plot shows m⁶A peaks detected in control mESCs within indicated genomic regions. Peak frequency graph shows the distribution of m⁶A peaks across transcripts, with a strong enrichment at the 3′ end. Genome browser tracks showing m⁶A distribution at the *Klf4* locus in mESCs. Red bars correspond to m⁶A-RIP-seq tracks. Gray bars

correspond to input tracks. **g** Correlation plot showing the log2 fold change of gene expression in RBM15 KO compared to ΔRRM1 mESCs. The linear regression line was fitted using Pearson correlation and *p*-value calculated using a two-sided *t*-test. All significantly (padj < 0.05) changed genes that overlap in both datasets are colored in blue (downregulated) and red (upregulated). **h** Correlation plot showing the log2 fold change of m⁶A peaks in RBM15 KO compared to ΔRRM1 mESCs. The linear regression line was fitted using Pearson correlation and *p*-value calculated using a two-sided *t*-test. All significantly (padj <0.05) changed peaks that overlap in both datasets are colored in cyan (downregulated) and magenta (upregulated). **i** Venn diagrams displaying the overlap between RBM15 KO and ΔRRM1 differentially regulated genes (mRNA-seq) and m⁶A peaks (m⁶A-RIP-seq). Statistical significance calculated using a two-sided Fisher's exact test. **j** Volcano plot showing differentially m⁶A modified mRNA sites in SWI/SNF genes between NT and *Rbm15* KO mESCs (*n* = 2, padj <0.05 and |FC| > 1.5). The downregulated peaks are in cyan and the upregulated peaks are in magenta. Adjusted *p*-values were calculated using the Benjamini-Hochberg correction using DESeq2 software, see Methods. **k** Table summarizes changes in mRNA levels and m⁶A deposition for all SWI/SNF genes when comparing RBM15 KO to NT mESCs. **l** Genome browser tracks showing m⁶A distribution at selected SWI/SNF subunit mRNAs in RBM15 KO and NT mESCs. Significantly increased m⁶A peaks in RBM15 KO cells are highlighted by magenta boxes and decreased peaks by cyan boxes. Source data are provided as a Source Data file.

requires the correct stoichiometry of subunits to form fully functional chromatin remodelers[9-11]. If certain subunits are more or less abundant, incomplete complexes are assembled, which can impair chromatin remodeling activity. To test this hypothesis, we performed western blot analysis of six SWI/SNF subunits in NT and RBM15 KO mESCs (Fig. 6b, c and Supplementary Fig. 6c). We found that certain subunits that had increased m⁶A mRNA levels also had increased protein levels (ex: SMARCC1, ARID1A), whereas other subunits showed no changes (ex: SMARCA4, SMARCD1). To determine the impact of altered subunit levels on SWI/SNF complex assembly, we performed MS1 based label-free quantitative immunoprecipitation mass-spectrometry (IP-MS) measurements of SWI/SNF complexes using V5-Trap nanobodies in NT and KO mESCs (Fig. 6d, Supplementary Fig. 6a and Supplementary Data 5). Using this sensitive proteomics method, we detected all subunits present in SWI/SNF complexes in mESCs. We found that SWI/SNF complexes purified from RBM15 KO mESCs were strongly enriched in core complex subunits relative to the catalytic ATPase/ARP subunits (Fig. 6e). Indeed, we measured a 2 to 4-fold increase in core subunits (ex: SMARCC1, SMARCE1 and SMARCB1) compared to the ATPase/ARP subunits (ex: SMARCA4, BCL7B, ACTINB). Previous studies have shown that SWI/SNF complexes assemble in a step-wise process[59]. First, a module containing SMARCC1/2 dimers, SMARCD1/2/3, SMARCE1 and SMARCB1 is formed. Next, the ARID1A/B and DPF1/2/3 module is added to form the core of the SWI/SNF complex. Finally, the catalytic ATPase subunit SMARCA2/4 and ARP subunits ACTL6A/B, ACTINB and BCL7A/B/C are added to assemble a fully functional SWI/SNF chromatin remodeling complex (Fig. 6f). Based on the modular organization of SWI/SNF complexes, our IP-MS data suggest that in RBM15 KO mESCs many SWI/SNF complexes fail to fully assemble, and they lack the ATPase/ARP module which is required for chromatin remodeling activity (Fig. 6f). To confirm that the regulation of SWI/SNF subunit abundance by the m⁶A-writer complex is dependent on RBM15 and not RBM15B, which is also expressed in mESCs (Supplementary Fig. 6b), we generated RBM15B KO *ZF-DT-Nkx2.9* mESCs (Supplementary Fig. 6d). In the SWI/SNF recruitment assay, we found that RBM15B KO did not rescue the DT-induced cell death after 48 h of ABA treatment (Supplementary Fig. 6e). Western blot analysis also revealed no changes in SWI/SNF subunit abundance in RBM15B KO cells compared to NT controls (Supplementary Fig. 6f). These data support the results of the genetic screen (Fig. 1e) which did not identify RBM15B as a regulator of SWI/SNF activity. Finally, there are over 400 genes that encode for

chromatin regulators (CRs) in mammals, many of which assemble into multi-subunit complexes similar to the SWI/SNF complex. We therefore hypothesized that m⁶A methylation may regulate the subunit stoichiometry of other CR complexes. To test this hypothesis, we evaluated changes in m⁶A levels of 326 CR genes[60] in RBM15 KO mESCs. We found that 91/326 CR genes displayed dynamic changes in m⁶A distribution on their mRNAs (Fig. 6g). In particular, subunits of the ISWI, SWI/SNF and CHD chromatin remodeling complexes were differentially methylated, as well as many members of the histone demethylase and histone methyltransferase families. To link the changes in m⁶A levels to changes in protein levels we performed western blot analysis for several chromatin regulators with increased m⁶A methylation in RBM15 KO mESCs. Indeed, we found that DNMT1 and ASH2L protein levels were increased in RBM15 KO cells compared to NT controls, whereas EZH2 and HDAC1 levels were unchanged (Supplementary Fig. 6g). This analysis suggests that m⁶A RNA methylation may be a general mechanism employed by cells to control subunit dosage and stoichiometry of mammalian chromatin regulator complexes.

## Discussion
Forward genetic screens in model organisms have played an instrumental role in the discovery and characterization of SWI/SNF chromatin remodelers[21-24]. However, mammalian SWI/SNF complexes have unique properties, such as an increase in subunit diversity that allows for cell-type specific functions during development[19]. CRISPR/Cas9-based methods now make it possible to perform genome-wide KO screens in mammalian cells[30], offering the opportunity to uncover new mechanisms that regulate SWI/SNF chromatin remodeling activity in mammals. To conduct a genome-wide loss-of-function screen in mESCs, we developed an innovative reporter assay using resistance to Diphtheria Toxin mediated cell death. Unlike previous studies that relied on fluorescence-activated cell sorting (FACS) for screening knock-out cells with transcription-based readouts, our method leverages the advantages of live/dead reporter assays[61]. By enriching for mutants without the need for cell sorting, live/dead selection typically achieves superior signal-to-noise ratios[62]. Our approach uses the cytotoxic DT-A gene as the reporter, enabling the identification of hits crucial for activating rather than silencing gene expression, as previously performed with antibiotic resistance reporter genes[62]. Additionally, we combined the DT-A live/dead reporter assay with epigenome editing tools to screen for regulators of SWI/SNF

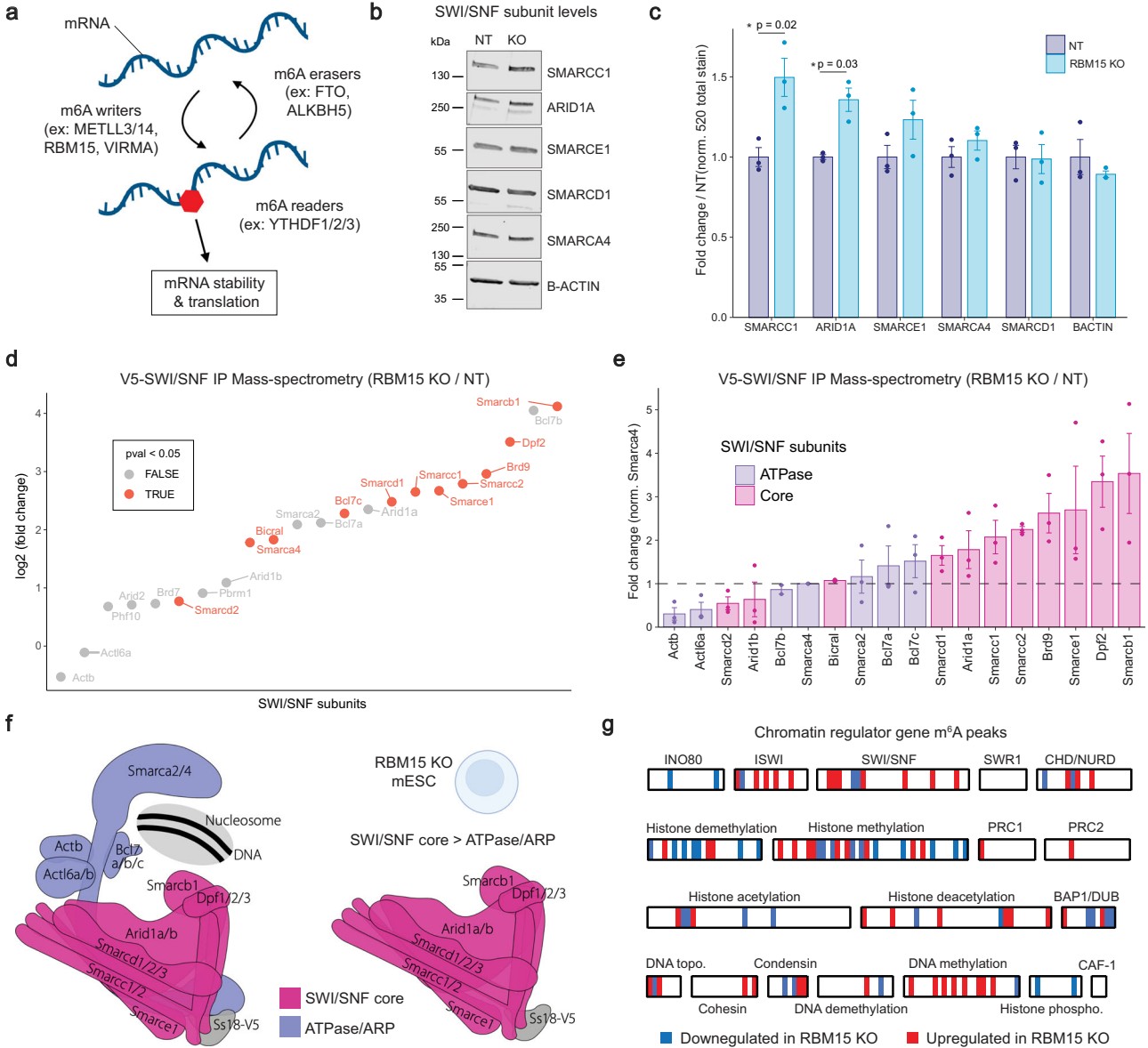

**Fig. 6 | RBM15 regulates SWI/SNF complex assembly in mESCs. a** Schematic of the m⁶A RNA methylation pathway that regulates mRNA stability and translation. Created in BioRender https://BioRender.com/houenb9. **b** Western blot analysis of SWI/SNF subunits in RBM15 KO and NT mESCs. (*n* = 3). Note: B-ACTIN is a subunit of the SWI/SNF complex. **c** Graph shows quantification of replicates normalized to total protein stain, depicted as mean ± SEM (*n* = 3). Significance calculated using two-sided *t*-test. **d** Graph shows relative abundance of SWI/SNF subunits in complexes immunoprecipitated from RBM15 KO and NT mESCs. SWI/SNF complexes were isolated via V5-tagged SS18 subunits and peptide abundance was determined by MS1 based label-free quantitative immunoprecipitation mass-spectrometry (*p*-val <0.05, *n* = 3). *P*-values were calculated using ANOVA test. **e** Graph shows enrichment of SWI/SNF subunits relative to the ATPase subunit SMARCA4 following V5-SWI/SNF IP-MS. Fold change calculated over SMARCA4 depicted as mean ± SEM (*n* = 3). **f** Schematic of SWI/SNF complexes isolated from RBM15 KO mESCs, core subunits of the complex are enriched relative to the ATPase/ARP subunits leading to impaired SWI/SNF activity. Created in BioRender https://BioRender.com/887opbj. **g** Heatmap shows dynamic changes in m⁶A levels on 96/379 chromatin regulator genes profiled in m⁶A-RIP-seq analysis of RBM15 KO mESCs. Source data are provided as a Source Data file.

chromatin remodeling activity. This revealed that SETD1B and TET1 function together with SWI/SNF complexes to establish an active epigenetic landscape at bivalent genes in mESCs. Our screening strategy extends beyond SWI/SNF complexes and bivalent genes in ES cells, offering a versatile method to uncover novel regulators of the dozens of chromatin regulators that promote gene expression across different cell types, including MLL1-4 histone methyltransferases. This approach is also ideal for chemical library screening to identify small molecules that modulate SWI/SNF activity and have high therapeutic potential.

We identify a new role for MLF2 as a regulator of SWI/SNF chromatin remodeling activity. We show that MLF2 interacts with SMARCA4 containing SWI/SNF complexes in the nucleus, and

regulates a subset of SWI/SNF-dependent genes by promoting chromatin accessibility at sites with high levels of SWI/SNF binding. In flies, MLF acts in a chaperone complex with HSP70 to regulate the stability of transcription factors and impact gene expression[46]. In our genetic screen we also identified *Hspa4*, a gene encoding for a member of the HSP70 chaperone family, suggesting that a similar chaperone mechanism regulates SWI/SNF complex assembly in mESCs. Since MLF2 does not modulate SWI/SNF activity by binding chromatin, it is likely that it has an upstream impact on the assembly of complexes. Indeed, we propose that MLF2 loss destabilizes SWI/SNF complex assembly, reducing its ability to bind chromatin. Since chromatin accessibility is only impacted by MLF2 degradation at strongly bound

SWI/SNF target sites, and MLF2 knockout only partially rescues the cell death phenotype in the SWI/SNF reporter assay, MLF2 is likely not essential for the stability of all SWI/SNF complexes but rather acts as a modulator of complex assembly. Thus, MLF2 represents a novel target for drugs aimed at reducing SWI/SNF activity without the toxicity associated with complete inhibition of chromatin remodeling activity. In line with this, previous studies have shown that MLF2 over-expression increases cell proliferation in various cancer cell lines, while MLF2 downregulation decreases proliferation[44] as observed in MLF2-dTAG mESCs. In addition to its role in cancer, MLF2 dysregulation is observed in neurological conditions such as DYT1 dystonia, a movement disorder caused by the formation of nuclear blebs containing high amounts of MLF2 and the HSP40-HSP70 chaperones[63].

One third of SWI/SNF genes are haploinsufficient (*SMARCA2/4*, *SMARCC1/2*, *ARID1A/B*, *SMARCD1*, *SMARCE1*, *SMARCB1* and *DPF2*), underscoring their dosage sensitive role in human development[10]. Indeed, many neurodevelopmental disorders and cancers arise from mutations that alter SWI/SNF subunit expression levels, leading to incorrect assemblies of the SWI/SNF complex[7,8]. Today, we know very little about the mechanisms that control the relative abundance of SWI/SNF complex subunits and ensure proper complex assembly in healthy cells. How do cells titer subunit levels to ensure correct stoichiometry of these multi-subunit complexes? Here, we reveal that m6A RNA methylation controls the stoichiometry of SWI/SNF complex subunits. Specifically, we found that RBM15, a subunit of the m6A-writer complex, regulates m6A methylation on specific SWI/SNF subunit mRNAs. In support of our findings, a previous study has shown that *RBM15* knockdown in HEK293T and Neuro2A cell lines increases SMARCC1 levels and this regulation is dependent on METLL3, the catalytic subunit of the m6A-writer complex[58]. Additionally, we show that RBM15 mediates mRNA target selection via its RNA recognition motifs (RRMs). Notably, we find that in RBM15 ΔRRM1 mutant mESCs, a discreet subset of mRNAs exhibits differential m6A levels compared to full RBM15 KO mESCs, suggesting that this domain contributes to the recognition of specific mRNAs. Current models propose a passive methylation of all accessible m6A methylation sites by the writer complex. This is supported by evidence that steric hindrance from the Exon Junction Complex can exclude the writer complex from all potential methylation sites on short exons[64–66]. Here we provide evidence that the RRM domains of RBM15 impose a further selection on such accessible m6A methylation sites.

Using m6A-RIP-seq, we show that RBM15 KO in mESCs alters m6A distribution on several SWI/SNF subunit mRNAs including SMARCC1, which leads to increased levels of core subunits and aberrant SWI/SNF complex assembly. Importantly, RBM15B is also expressed in mESCs (Supplementary Fig. 6b) and this subunit competes with RBM15 for incorporation into the m6A-writer complex. Our data suggest that RBM15 and RBM15B have different roles in mESCs, most likely due to their unique sets of RNA recognition motifs. Indeed, unlike RBM15, RBM15B was not enriched in the genetic screen and its loss did not impact SWI/SNF activity or subunit abundance in mESCs. We also identified the *Ythdf1*, *Ythdf2* and *Ythdf3* m6A-readers in the genome-wide screen for SWI/SNF regulators. These genes have previously been shown to regulate both the stability and translation efficiency of m6A-methylated mRNAs[52]. Consequently, the absence of these m6A-readers may also alter the abundance of specific SWI/SNF subunit mRNAs and impair complex assembly, or the observed phenotype may act through a distinct mechanism as YTHDF1/2/3 readers have been shown to impact global m6A regulation. Notably, we also identified several pBAF subunit mRNAs with altered m6A-distributions in RBM15 KO mESCs, however we did not evaluate pBAF complex assemblies in our proteomics studies as we performed immunoprecipitation experiments via the SS18 subunit which is not present in pBAF.

Finally, many genes identified in our screen are expressed in the nervous system and several have been found mutated in patients with neurodevelopmental disorders[67]. Our analysis of MLF2 target genes in mESCs suggests that MLF2 may also have a role in neural development that remains to be investigated. Interestingly, previous research in the developing mouse brain showed that overexpression of RBM15 impairs neurogenesis and this is phenocopied by *Smarcc1* knockout, thereby associating RBM15 with SWI/SNF activity in brain development[58]. Furthermore, the brain specific deletion of the m6A writer complex component METTL14 impairs cortical neurogenesis in mice[68]. The defects in METTL14 KO neural stem cells are associated with misregulation of histone methylation levels via destabilization of transcripts coding for histone-modifying enzymes. Our data suggests that the m6A RNA methylation pathway controls chromatin remodeler transcript abundance and complex assembly in mouse embryonic stem cells. Interestingly, many SWI/SNF mRNAs are also m6A-modified in human embryonic stem cells, with numerous SWI/SNF subunits displaying similar patterns of m6A modifications in mice and human ESCs (Supplementary Fig. 7). Previous studies have shown that m6A RNA methylation controls transcription via regulation of active histone modifications, heterochromatin and chromatin associated RNAs[69–72]. Here we reveal a crosstalk between m6A RNA methylation and chromatin remodeling in the regulation of gene expression. Given the essential function of chromatin remodelers, small molecule inhibitors targeting these proteins have shown limited efficacy in clinical trials due to high cytotoxicity. Consequently, targeting modulators of chromatin remodeling activity such as RBM15 and MLF2 may offer potent, yet less toxic, therapeutic strategies for diseases associated with impaired chromatin regulation.

## Methods

### Mouse embryonic stem cell (mESC) cultures

Wild-type mESCs derived from blastocysts of mixed 129-C57Bl/6 background were cultured in media containing DMEM (Gibco, 31966-021), supplemented with LIF (in-house production), 15% FBS (Gibco, 10437-028), 0.1 mM β-Mercapthoethanol (Gibco, 31350010), 1x non-essential amino acids (Gibco, 11140035), 100 mM Hepes buffer (Gibco, 15630056), 100 U/ml Penicillin-Streptomycin (Gibco, 15140122) at 37 °C and 5% $CO_2$. All cell culture medium was filtered using Stericup Filter system (Millipore, S2GPU05RE). Cells were cultured on plates coated for 30 min with 0.2% gelatin solution (Sigma, G1890-100G) and passaged every two to three days by dissociation with 0.05% Trypsin-EDTA (Gibco, 25300054). For SWI/SNF-recruitment cells were treated with 0.1 mM ABA (Sigma-Aldrich, 90769-25MG) for 48 h and an equal volume of EtOH as control. For dTAG-13 induced protein degradation cells were treated with 500 nM dTAG-13 (Sigma-Aldrich, SML2601-5MG) or an equivalent volume of DMSO (Sigma-Aldrich, D2650-100ML) for 3 h, 8 h, or 24 h. For METLL3 inhibition, mESCs were treated with 1 uM STM2457 (MCE, HY-134836) for 48 h prior to ABA treatment. mESCs were kept in 1 uM STM2457 during ABA treatment. RBM15 KO and ΔRRM1 mutant mESCs displayed normal proliferation rates and colony morphology when cultured in the presence of MEF-feeders (Supplementary Fig. 3e). However, RBM15 KO and ΔRRM1 mutant mESCs began to differentiate after prolonged culture in feeder-free conditions and therefore all experiments were performed at early passages after MEF withdrawal.

### Plasmid cloning

SWI/SNF recruitment fusion proteins were cloned into a piggyBac plasmid (pSB118 EF1a PYL1-SS18 PGK ABI-ZFHD1 2 A Blast) using In-Fusion HD cloning (Takara, 638947) with DNA sequences amplified by PCR from Addgene plasmids (102881, 44017, 84239). For the *ZF-DT-Nkx2.9* mESC line, ZFHD1 binding motifs were knocked into the *Nkx2.9* locus using the CiA cassette[28] that was cloned into a homology directed repair (HDR) template plasmid with 0.5 kb homology arms flanking the *Nkx2-9* promoter. The DT-A reporter gene was cloned into the start codon of the *Nkx2-9* gene to generate an HDR template with 0.5-1 kb homology arms. For the dTAG mESC lines, the FKBPv-V5 tag was

obtained from Addgene plasmid 91798 and knocked into the *Mlf2* and *Smarca4* loci using HDR templates containing 0.5-1 kb homology arms flanking the C-terminal stop codons. The genomic sequences for HDR templates were ordered as gene fragments (Twist Bioscience) using the mm10 genome. The sgRNAs were cloned into Addgene plasmid 62988 that also expresses Cas9, sequences are provided in Supplementary Data 6.

## CRISPR/Cas9 genome editing

mESCs were plated on gelatin coated plates with antibiotic resistant DR4 mouse embryonic fibroblasts (MEFs). For CRISPR/Cas9 knock-in (KI) a transfection mix containing 4 µg template DNA, 8 µg sgRNA/Cas9 plasmid (Addgene 62988, Supplementary Data 6), 200 µl Opti-Mem (Gibco, 11058021) and 25 µl Lipofectamine 2000 (Invitrogen, 11668019) was prepared. For CRISPR/Cas9 knockout (KO) a transfection mix containing 4 µg sgRNAa/Cas9 plasmid, 4 µg sgRNAb/Cas9 plasmid (Addgene 62988, Supplementary Data 6), 200 µl Opti-Mem and 25 µl Lipofectamine 2000 was prepared. The transfection mix was incubated for 15 min at RT before adding it to the cells. After 4 h, cell culture media was renewed. After 24 h, cells were put into 1.5 µg/ml Puromycin (InvivoGen, ANT-PR-1) selection for 48 h. Single colonies were manually picked after 3 days, dissociated with trypsin and expanded on MEFs. Feeder free cultured lines were expanded for DNA and protein extraction. mESC lines containing homozygous insertions were confirmed by PCR, Sanger sequencing and Western blotting.

## piggyBac stable cell lines

mESCs were plated on gelatin coated plates with MEFs and transfected with a mix of 5 µg pSB118 plasmid (expressing SS18-PYL1 and ZF-ABI1 fusion proteins), 2 µg piggyBac transposase plasmid, 200 µl Optimem (Gibco, 11058021) and 25 µl Lipofectamine 2000 (Invitrogen, 11668019). The transfection mix was incubated for 15 min at RT before adding it to the cells. After 4 h, cell culture media was renewed. After 48 h, cells were put into 10 µg/ml Blasticidin (InvivoGen, ANT-BL-1) selection for 4 days. Next, single colonies were picked to generate clonal cell lines. Successful transposition and stable expression of fusion proteins of mESC clones was confirmed via western blot.

## Western blot

Cell pellets were lysed in RIPA lysis buffer (10 mM Tris-HCl pH8, 150 mM NaCl, 1 mM EDTA pH8, 0.5 mM EGTA pH8, 1% NP-40, 0.5% DOC, 0.1% SDS) for 20 min at 4 °C. Lysates were centrifuged at 4 °C, 13.000 rcf for 10 min and supernatant transferred to new tubes. Protein concentration was determined via BCA protein assay (Pierce, 23225), according to the manufacturer's instructions. Samples were mixed with 1x Loading buffer (stock at 5x: 10% SDS, 0.2 M Tris HCl pH6.8, 30% Glycerol, traces of Bromophenol blue), 20x DTT (1 M) and boiled for 5 min at 95 °C. Samples were loaded onto 4-15% Mini-protean TBX gel (BioRad, 4561084) and run for 1 h at 150 V in Running buffer (0.0125 M Tris, 0.096 M Glycine). Transfer was conducted using the Trans-Blot Turbo RTA kit (BioRad, 1704270) according to the manufacturer's instructions. For total protein staining, Revert 520 total protein stain kit (LICORbio, 926-10010) was used according to the manufacturer's instructions. Alternatively, membranes were stained with Ponceau-S solution (0.1% Ponceau red dye, 5% Acetic acid). Next, membranes were blocked for 1 h in 5% Milk in TBS-T (0.02 M Tris, 0.15 M NaCl, pH7.6 adjusted with HCl, 0.1% Tween20). Primary Antibody (Supplementary Data 6) was added in appropriate dilutions in 5% Milk in TBS-T and incubated over night at 4 °C. Membranes were washed three times for 10 min in TBS-T, before incubation with secondary antibody, diluted 1:10'000 in 5% Milk in TBS-T (Supplementary Data 6) for 1 h at RT. Membranes were again washed three times for 10 min in TBS-T. Membranes were imaged on an Odyssey DLx infrared imager (LICORbio). Western blot quantification was done using Empiria studio software (LICORbio).

## Sequential salt extraction

Sequential salt extraction was performed as described by Porter et al.[73]. In brief, 8 million cells were collected and resuspended in 1 ml Buffer A (25 mM HEPES pH 7.6, 25 mM KCl, 5 mM MgCl$_2$, 0.05 mM EDTA, 0.1% NP-40, 10% glycerol) and incubated for 10 min at 4 °C with overhead rotation. Lysates were centrifuged (5 min, 6000 g, 4 °C) and supernatants collected (cytosolic fraction). Next, the pellet was resuspended in 200 µl mRIPA buffer (100 mM Tris pH 8.0, 2% NP-40, 0.5% sodium deoxycholate) by pipetting 15 times. The lysate was incubated for 3 min on ice, centrifuged (3 min, 6500 g, 4 °C) and supernatant collected (fraction 1). This process was repeated with mRIPA buffer of increasing NaCl concentrations. Finally, 40 µl of each fraction was processed for western blot analysis.

## Proximity ligation assay

MLF2-dTAG mESCs were seeded on to gelatin coated coverslips and treated with DMSO or dTAG-13 for 24 h. Cells were fixed in 4% PFA, washed in PBS and permeabilized in 0.1% Triton for 15 min. The Duolink in situ proximity ligation assay was then performed according to manufacturer's protocol (Duolink, Sigma-Aldrich, DUO92101). In brief, coverslips were incubated in Duolink blocking solution for 60 min at 37 °C then with mouse anti-V5 and rabbit anti-SMARCA4 primary antibodies diluted in Duolink Antibody Diluent overnight at 4 °C. Coverslips were then washed 2x before incubation with +/− PLA probes for 1 h at 37 °C. After 2x wash, the coverslips were incubated with a Ligation mix for 30 min at 37 °C, followed by 2x wash and final Polymerase amplification step for 100 min at 37 °C. Lastly, coverslips were washed 3x before mounting in Duolink mounting media with DAPI (n = 3 for each condition). Coverslips were imaged on a Stellaris 5 microscope (Leica) at 60x magnification. For PLA image quantification, we used ImageJ software according to the method by López-Cano et al. [74].

## Chromatin immunoprecipitation

Cells were trypsinized for 5 min, washed with DPBS, counted (20 million cells), and fixed for 30 min in 2 mM disuccinimidyl glutarate (DSG) followed by 12 min in 1% PFA. Crosslinking was then quenched with 0.125 M glycine and cells were incubated on ice for 5 min. Crosslinked cells were spun at 800 rcf for 5 min. Nuclei were prepared with 10 mL cell lysis buffer (50 mM HEPES pH 8.0;140 mM NaCl; 1 mM EDTA; 10% glycerol; 0.5% NP40; 0.25% TritonX100), then washed in 10 mL rinse buffer (10 mM Tris pH 8.0; 1 mM EDTA; 0.5 mM EGTA;200 mM NaCl). The chromatin pellets were resuspended in 900 µL shearing buffer (0.1% SDS, 1 mM EDTA pH 8.0, and 10 mM Tris pH 8.0) and sonicated in a Bioruptor Pico sonicator (Diagenode) for 4 x 30 s cycles. Sonicated chromatin was centrifuged at 10,000 rcf for 5 min, and the supernatant was collected. 25 uL input chromatin was stored at 4 °C and the IP reactions were setup as follows: 250 uL of sonicated chromatin was diluted with 0.25 volume of 5xIP buffer (250 mM HEPES, 1.5 M NaCl, 5 mM EDTA pH8.0, 5% Triton X-100, 0.5% DOC, and 0.5% SDS) and incubated for 12−16 h at 4 °C with 25 µL protein G Dynabeads (Invitrogen, 10004D) and 5 µg of antibody (Supplementary Data 6). The beads were then washed three times with 1 mL 1 x IP buffer, once with 1 mL DOC buffer (10 mM Tris pH 8; 0.25 M LiCl; 0.5% NP40; 0.5% DOC; 1 mM EDTA), and then eluted in 150 µL elution buffer (TE buffer with Proteinase K and 1% SDS). After reverse crosslinking overnight, the ChIP DNA was purified using MinElute clean-up columns (Qiagen, 28004) and resuspended in 30 µL EB solution. Sequencing libraries were prepared from 20 uL of ChIP DNA using the NEBNext Ultra II DNA Library Prep Kit for Illumina (NEB, E7645S). Library molarity and quality were assessed by Qubit and Tapestation (DNA High sensitivity chip). Libraries were sequenced on a NovaSeq 6000 Illumina sequencer with paired-end 50 settings.

## Lentivirus preparation

GeCKOv2 pooled sgRNA lentivirus libraries A and B[30] were produced for infection of the CRISPR screen reporter *ZF-DT-Nkx2.9* mES cell line. For lentivirus production HEK293T cells were seeded on 15 cm cell culture plates in DMEM media (Gibco, 31966-021), supplemented with 10% FBS (Gibco, 10437-028) and 100 U/ml Penicillin-Streptomycin (Gibco, 15140122). Cells were cultured at 5% $CO_2$ and 37 °C. For transfection, a transfection mix containing 10.8 ml Opti-Mem (Gibco, 11058021), 840 µl PEI (Polysciences, 24765), 27 µg plasmid pMD2.G (Addgene 12259), 81 µg plasmid psPAX2 (Addgene 12260) and 108 µg of library DNA was mixed and incubated for 10 min at RT. On each cell culture plate 1.8 ml of transfection mix was added. The following day the media was changed to mESC media – LIF (recipe see mESC cell cultures). Two days later, the supernatant of all plates was collected and centrifuged at 1000 g for 5 min. The supernatant was filtered and the collected virus was stored at 4 °C for infection of mESCs.

## CRISPR KO screen

50 million *ZF-DT-Nkx2.9* mESCs were plated for each library. The next day, cells were transduced with GeCKOv2 lentivirus libraries A and B[30] at an MOI of <0.4. 24 h after transduction cells were split into mESC media. 48 h after transduction, 1.5 µg/ml Puromycin (InvivoGen, ANT-PR-1) and 10 µg/ml Blasticidin (InvivoGen, ANT-BL-1) were added to the cell culture media. Cells were kept in selection for 5 days. Importantly, cell number was always kept at a minimum of 100 mio cells/library. After selection 100 mio cells/library were pelleted and stored at −20 °C. At the same time 100 mio cells/library were plated into 0.1 mM ABA (Sigma-Aldrich, 90769-25MG). After 48 h treatment with ABA, surviving cells were collected and stored at −20 °C. To identify enriched sgRNAs we followed a published protocol[75]. A QiaAMP DNA isolation kit (Qiagen, 51192) was used to isolate the DNA of all collected cells. Isolated DNA was Concentrated: 0.1 Volumes 3 M Sodium acetate (pH 5.2) and three volumes of ice cold 100% EtOH were added to the DNA samples. Samples were incubated at −20 °C for 4 h. Next, samples were centrifuged (13,000 rpm, 4 °C, 30 min) and the pellets washed twice with 0.5 ml ice cold EtOH; spinning for 10 min each time. EtOH was removed and pellets air dried for 15 min before resuspending in 500 µl $H_2O$. To amplify the inserted sgRNA sequences a first PCR was performed using NEBNext Ultra II Q5 Master Mix (M5044S) and GeCKO F1 and R1 primers (Supplementary Data 6). PCR products were cleaned up using the MinElute kit (Qiagen, 28004) and DNA was eluted in 50 µl $H_2O$. Next, a second PCR was performed to add sequencing barcodes to the amplified sgRNAs. For each condition a different primer pair was used (Supplementary Data 6). PCR products were loaded on a 1.2% Agarose gel. The resulting band at about ~250 bp was cut out and DNA was purified using a PCR & Gel clean up kit (Qiagen, 28506).

## RNA-seq

For RNA-seq analysis cell pellets were collected and stored at −80 °C. RNA was then isolated using the RNAeasy kit (Qiagen, 74106). Sequencing libraries were generated using the Collibri 3′ mRNA Library Prep Kit for Illumina (Invitrogen, A38110024), according to the manufacturer's instructions. Libraries were sequenced on an NovaSeq 6000 Illumina sequencer with single-end 100 settings.

## ATAC-seq

For ATAC-seq[76] 60.000 cells were collected and resuspended in 500 µl RSB buffer (10 mM Tris-HCl pH7.4, 10 mM NaCl, 3 mM $MgCl_2$). Cells were pelleted at 500 rcf, at 4 °C for 5 min. Supernatant was removed, cells were resuspended in 50 µl RSB buffer containing 0.1% Tween-20 and 0.1% NP40 and incubated on ice for 3 min. Lysis was washed out with 500 µl RSB buffer containing 0.1% Tween-20. Samples were centrifuged for 10 min at 4 °C and 500 rcf. Supernatant was removed and pellets resuspended in 50 µl transposition mix (25 µl TD buffer (Diagenode, C01019043-1000), 2.5 µl Tagmentase (Diagenode, C01070012-

200), 22.5 µl $H_2O$). Samples were incubated for 30 min at 37 °C at 1000 rpm. DNA was cleaned up using the MinElute kit (Qiagen, 28004). Eluted DNA was amplified for 10 cycles, adding Nextera Ad PCR primers. DNA was cleaned up using the MinElute kit (Qiagen, 28004). Library molarity and quality were assessed by Qubit and Tapestation (DNA High sensitivity chip). Libraries were sequenced on a NovaSeq 6000 Illumina sequencer with paired-end 50 settings.

## Cut & Run

For CUT&RUN analysis, 500'000 cells were collected and directly processed. The experiment was conducted with the CUTANA ChIC/CUT&RUN Kit (Epicypher, 14-1048), according to the manufacturer instructions. Sequencing libraries were prepared from 2 ng/µL of CUT&RUN-enriched DNA using the Illumina TruSeq ChIP Library Preparation Kit and the IDT TruSeq RNA UD Indexes v2. No size selection was performed. Library molarity and quality were assessed by Qubit and Tapestation (DNA High sensitivity chip). Libraries were sequenced on a NovaSeq 6000 Illumina sequencer with paired-end 50 settings.

## m⁶A-RIP-seq

For m⁶A-RIP-seq[57] 15 mio cells were collected and stored at −80 °C. RNA was isolated using the RNAeasy kit (Qiagen, 74106). Next, mRNA was isolated using the mRNA Isolation kit (NEB, S1550S). mRNA was fragmented using the NEBNext Magnesium RNA fragmentation module (NEB, E6150S) and purified using Monarch RNA Cleanup Kit (NEB, T2030S). 100 ng mRNA was aliquoted and stored as input at −80 °C. For m⁶A-RIP, 50 µl of Dynabeads protein G (Invitrogen, 10004D) were aliquoted and placed on a magnet. Supernatant was removed and beads were washed in 500 µl of m⁶A-IP buffer (10 mM TrisHCl pH7.4, 150 mM NaCl, 0.1% NP-40). After the initial washing, beads were resuspended in 500 µl m⁶A-IP buffer and incubated for 45 min at RT on a rotating wheel. Next, supernatant was removed and beads were resuspended in 400 µl m⁶A-IP buffer. Samples were placed on ice and 5 µg of m⁶A-antibody (Supplementary Data 6), 5 µl RiboLock RNase Inhibitor (Thermo Fisher Scientific, E00381), 20 µl fragmented mRNA, 100 µl 5X m⁶A-IP buffer (50 mM Tris-HCl pH7.4, 750 mM NaCl, 0.5% NP-40) and 370 µl $H_2O$ were added. Samples were incubated for 2 h, at 4 °C on a rotating wheel. Next, samples were placed on a magnet and supernatant was removed. Samples were washed twice in 500 µl ice cold m⁶A-IP buffer for 30 s. After washing samples were resuspended in 150 µl RNA cleanup binding buffer of the Monarch RNA Cleanup Kit (NEB, T2030S). Samples were incubated for 5 min at RT on a rotating wheel. The eluate was transferred to a new tube. m⁶A-RIP mRNA was purified using the Monarch RNA Cleanup Kit (NEB, T2030S). Samples were eluted in 8 µl elution buffer. The sequencing libraries were made using the Illumina Stranded mRNA Prep Ligation kit and protocol starting from 100 ng of input mRNA and m⁶A-RIP mRNA. To start, 8.5 ul of RNA, 8.5 ul of Fragmentation master mix and 8 ul of First Strand Synthesis master mix were mixed for direct First strand cDNA synthesis (no extra fragmentation step). Then protocol instructions were followed as described. Library molarity and quality were assessed by Qubit and Tapestation (DNA High sensitivity chip). Libraries were sequenced on a NovaSeq 6000 Illumina sequencer with paired-end 50 settings.

## Immunoprecipitation of SWI/SNF complexes

For each replicate 20 mio cells were collected. Cell pellets were resuspended in 500 µl ice cold EBO lysis buffer (50 mM Tris-HCl pH7.5, 0.05% NP-40, 1 mM EDTA, 1 mM $MgCl_2$), containing 1000x DTT (1 M) and Protease Inhibitor Cocktail (Roche, COEDTAF-RO). Cells were lysed for 5 min on ice, before centrifugation at 1000 rcf, 4 °C for 5 min. Supernatant was removed and pellets resuspended in 270 µl EB300 lysis buffer (50 mM Tris-HCl pH 7.5, 300 mM NaCl, 1% NP-40, 1 mM EDTA, 1 mM $MgCl_2$), containing 1000x DTT (1 M), Protease Inhibitor Cocktail (Roche, COEDTAF-RO) and 0.04 mg/ml DNaseI. Samples were

incubated for 10 min on ice with gentle vortexing every 3 min. Next, samples were centrifuged for 10 min, at 21,000 rcf, at 4 °C. 250 μl of supernatant was mixed with 250 μl dilution buffer (50 mM Tris-HCl pH7.5, 1 mM EDTA), containing Protease Inhibitor Cocktail (Roche, COEDTAF-RO). After extraction of nuclear proteins, 25 μl of V5-Trap magnetic beads (chromoTek, M-270) were washed three times with ice cold wash buffer (50 mM Tris-HCl pH7.5, 150 mM NaCl, 0.1% NP-40, 1 mM EDTA). Next, the diluted lysates were added to the beads and incubated overnight at 4 °C on an end-to-end rotator. The next day, beads were separated on a magnet to discard the supernatant. Beads were washed with 500 μl wash buffer three times, before resuspension in 50 μl DPBS (Gibco, 14190-094). Samples were further processed for Mass spectrometry.

## MS1 based label free quantitative IP-MS

MS analysis was always performed in triplicates for each condition. Immunoprecipitated samples were treated using iST kits (Preomics) according to the manufacturer's instructions. Briefly, beads were re-suspended in 50 μl of provided lysis buffer and proteins were denatured, reduced and alkylated during 10 min at 60 °C. The resulting slurries (beads and lysis buffer) were transferred to dedicated cartridges and proteins were digested with a Trypsin/LysC mix for 2 h at 37 °C. After two cartridge washes, peptides were eluted with 2 × 100 μL of provided elution buffer. Samples were finally completely dried under speed vacuum and stored at −20 °C. For MS analysis, samples were dissolved in 20 μl of loading buffer (5% CH3CN, 0.1% FA) and 4 μl were injected on column. LC-ESI-MSMS was performed on an Orbitrap Fusion Lumos Tribrid mass spectrometer (Thermo Fisher Scientific) equipped with Vanquish NEO liquid chromatography system (Thermo Fisher Scientific). Peptides were trapped on a PepMap NEO, C18, 5 μm, 300 μm x 5 mm trap cartridge (Thermo Fisher Scientific) and separated on a 75 μm x 500 mm, C18 ReproSil-Pur (Dr. Maisch GmBH), 1.9 μm, 100 Å, home-made column. The analytical separation was run for 85 min using a gradient of H$_2$O/FA 99.9%/0.1% (solvent A) and CH$_3$CN/FA 80%/0.1% (solvent B). The gradient was run from 95 % A and 5% B to 65% A and 35% B in 60 min, and then to 1% A and 99% B in 10 min with a pause for 15 min at this composition. Flow rate was 250 nL/min. Data-dependant analysis (DDA) was performed with MS1 full scan at a resolution of 120'000 FWHM followed by as many subsequent MS2 scans on selected precursors as possible within 3 s maximum cycle time. MS1 was performed in the Orbitrap with an AGC target of 4 × 105, a maximum injection time of 50 ms and a scan range from 400 to 2000 m/z. MS2 was performed in the ion-trap with an AGC target at 1 × 104 and a maximum injection time of 35 ms. Isolation windows was set at 1.2 m/z and 30% normalised collision energy was used for higher-energy collisional dissociation (HCD). Dynamic exclusion was set to 20 s.

## qPCR

DNA from ATAC and ChIP experiments was processed for qPCR analysis. 1 μl of DNA was mixed with 10 μl of SYBR green (applied Biosystems, A25742), 5 μl of H$_2$O and 4 μl of forward and reverse primers at 2.5 μM (Supplementary Data 6). The 96 qPCR well plates were measured on a QuantStudio1 thermocycler (Applied Biosystems).

## Immunostaining of cells

Cells were grown on cover slips. Cell culture media was removed and cells were washed with DPBS. 4% PFA was added to the wells and cells were fixed for 15 min at RT. Cells were washed with DPBS for 5 min at RT, before blocking for 1 h at RT in DPBS+ (5% Goat serum, 0.1% Triton). Next, primary antibodies were added at appropriate dilutions in DPBS+ (Supplementary Data 6) and incubated over night at 4 °C. The next day, cells were washed three times in DPBS at RT. Secondary antibodies were added at appropriate dilutions in DPBS+ (Supplementary Data 6) and incubated for 1 h at RT. Cells were again washed for three times with

DPBS at RT, mounted onto slides using DAPI Fluoromount-G (Southern Biotech, 0100-20) and imaged at the Stellaris 5 microscope (Leica).

## Cell survival assay

*ZF-DT-Nkx2.9* mESCs and derived CRISPR/Cas9 KO cell lines were plated into 0.1 mM ABA (Sigma-Aldrich, 90769-25MG) and an equal Volume of EtOH as control. After 48 h cells were trypsinized and resuspended in mESC medium. 10 μl of cell suspension was mixed with 10 μl Trypan blue (Gibco, 15250061). Cell number was measured at the TC20 automatic cell counter (Biorad). An average of two counts/samples was taken. Fold change was calculated by dividing the number of ABA treated cells by the number of EtOH treated cells. For Giemsa staining, cells were fixed with MetOH for 5 min at RT. Fixed cells were washed with DPBS (Gibco, 14190-094), air dried for 5 min and stained with Giemsa solution (Sigma-Aldrich, 48900-500ML-F) for 5 min. After washing with H$_2$0 the plates were scanned on the Cytation 5 plate imager (BioTek).

## ATPlite cell viability assay

Cells were seeded in a 96-well plate. At the desired time point 50 μl of ATPlite reagent (PerkinElmer, 6016943) was added to each well containing 50 μl of mESC medium. The plate was shaken at 700 rcf for 2 min before brief centrifugation. Plates were measured at the SpectraMax L 384w (Molecular devices).

## CRISPR screen MAGeCK analysis

Sequencing library molarity and quality were assessed by Qubit and Tapestation (DNA High sensitivity chip). 12 ul of Read 1 custom primer (Supplementary Data 6) at 100 uM were added to Illumina Read 1 primers (position 24 on NovaSeq cluster cartridge) before sequencing with paired-end 50 settings. To map reads to corresponding sgRNAs and target genes we used the software MAGeCK[31] v0.5.9.5 and determined significantly enriched genes using default normalization settings. We applied an Enrichment positive score cut off of 1.00E-4 to identify hits. Output data were plotted with ggplot2.

## RNA-seq analysis

Sequencing reads were trimmed using cutadapt[77] v3.5 (removal of truseq adapter sequences, polyA sequences, and low-quality reads). Trimmed reads were aligned to the mouse genome (mm10) using STAR[78] v2.7.5 (STARoptions, --readFilesCommand zcat --runThreadN 8 --outFilterMultimapNmax 20 --alignSJoverhangMin 8 --alignSJDBoverhangMin 1 --outFilterMismatchNmax 999 --outFilterMismatch NoverLmax 0.6 --alignIntronMin 20 --alignIntronMax 1000000 --alignMatesGapMax 1000000). Gene count matrices were obtained from the aligned reads using featureCounts[79] v2.0.3 (-t exon -g gene_id -O -T 4 -s 1 -a GRCm38.102). Differential gene expression analysis between conditions was determined using the R package DESeq2[80] after pre-filtering genes with low counts (rowSums > 10), with significance cut-offs set at FDR < 0.05 and fold change > 1.5 in either direction. Output data were plotted with ggplot2.

## m$^6$A-RIP-seq analysis

Sequencing reads were trimmed using cutadapt 3.5 (adapters -m 20 -O 4 -e 0.2 -p). Reads were aligned to the mm10 genome using STAR v2.7.5 (--readFilesCommand zcat --runThreadN 6 --genomeDir GRCm38.102 --outSAMtype BAM SortedByCoordinate --outSAMunmapped Within --outSAMattributes Standard). m$^6$A peaks enriched relative to input mRNA were called using MACS2[81] v2.7.1 (callpeak --nomodel --keep-dup all -B -q 0.05 --extsize 100 --shift −50 -g mm -t m$^6$A.bam -c input.bam, filtered for score > 200). Peaks from NT, KO and ΔRRM1 datasets within 100 bp were merged using bedtools v2.30 and the numbers of reads from each replicate overlapping with the peak set was counted using featureCounts v2.0.3 (-T 4 -O -p -a -t exon -g gene_id). Differential m$^6$A peak calls were determined using the R software DESeq2 after pre-filtering peaks with low counts (rowSums > 50), with

significance cut-offs set at padj <0.05 and fold change > 1.5 in either direction. Output data were plotted with ggplot2. For IGV genome browser snapshots, average of replicates genome coverage files (bigwig) were generated with deepTools[82] 3.5.2 (bamCoverage --binSize 10 --normalizeUsing RPKM and bigwigAverage for merging replicates). Genomic features of m⁶A peaks and the distribution along mRNA transcripts was determined using the R software ChIPSeeker[83] v1.32.1.

### ChIP-seq analysis
Sequencing reads were trimmed using cutadapt 3.5 (adapters -m 20 -O 5 -p). Reads were aligned to the mm10 genome using bowtie2 v2.4.4 (--sensitive -p 8). Alignments were filtered using samtools[84] v1.12 (view -h -F 1796 -q 20). V5-ChIP peaks enriched relative to inputs were called using MACS2 v2.7.1 (callpeak,--format BAMPE -g mm), however <100 peaks were called in all replicates, suggesting no enrichment relative to input. For IGV genome browser snapshots, genome coverage files (bigwig) were generated with deepTools 3.5.2 (bamCoverage --binSize 10 --normalizeUsing RPKM).

### ATAC-seq analysis
Sequencing reads were trimmed using cutadapt 3.5 (adapters -m 20 -O 5 -p). Reads were aligned to the mm10 genome using bowtie2 v2.4.4 (--maxins 2000 -p 8 -N 1). Alignments were filtered using samtools v1.12 (view -h -F 1796 -q 20). ATAC peaks were called using MACS2 v2.7.1 (callpeak --format BAMPE --call-summits --shift −75 --extsize 150 --keep-dup all -B --SPMR -g mm -t ATAC.bam). Peaks from all replicates within 1 kb were merged using bedtools v2.30, filtered against the mouse blacklist, and the numbers of reads from each replicate overlapping with the peak set was counted using featureCounts v2.0.3 (-T 4 -O -p -a -t exon -g gene_id). Differential ATAC peak calls were determined using the R software DESeq2 after pre-filtering peaks with low counts (rowSums > 100), with significance cut-offs set at FDR < 0.05 and fold change > 1.5 in either direction. Output data were plotted with ggplot2. For IGV genome browser snapshots, average of replicates genome coverage files (bigwig) was generated with deepTools v3.5.2 (bamCoverage --binSize 10 --normalizeUsing RPKM --ignoreForNormalization chrM and bigwigAverage for merging replicates). Genomic features of ATAC peaks were determined using the R software ChIPSeeker v1.32.1. deepTools v3.5.2 was used to produce heat map of published SMARCA4 ChIP-seq mean read density across ATAC. GREAT software was used to annotate ATAC peaks with genes to determine gene ontology terms associated with changes in accessibility.

### CUT&RUN analysis
Sequencing reads were trimmed using cutadapt 3.5 (adapters -m 20 -O 5 -p). Reads were aligned to the mouse mm10 and E. coli K12 MG1655 genomes using bowtie2 v2.4.4 (-p 8 --very-sensitive --no-unal --no-mixed --no-discordant --dovetail -X 1000). Alignments were filtered using samtools v1.12 (view -h -F 1796 -q 20). CUT&RUN peaks in mm10 were called using MACS2 v2.7.1 (callpeak --format BAMPE --nomodel --call-summits --shift −100 --extsize 200 --keep-dup all -B -g mm). Peaks from all replicates within 1 kb were merged using bedtools v2.30. Normalization scaling factors were calculated based on the spiked-in E. coli reads obtained from each replicate. Genome coverage files (bigwig) were generated with deepTools v3.5.2 (bamCoverage --binSize 10 --scaleFactor --normalizeUsing RPKM --ignoreForNormalization chrM ChrUn ChrRandom -p max). For IGV genome browser snapshots, average of replicates genome coverage files (bigwig) was generated with deepTools v3.5.2 (bigwigAverage for merging replicates). deepTools v3.5.2 was used to produce heat maps of SMARCA4 CUT&RUN mean read density across CUT&RUN and ATAC-seq peaks.

### Proteomics analysis
Raw data were processed using Proteome Discoverer 2.4 software (Thermo Fisher Scientific). Briefly, spectra were extracted and searched against the *Mus musculus* reference database (Uniprot, 17109 entries) with the native SS18 protein by sequence replaced the bait SS18-V5 protein sequence (used to purify SWI/SNF and ncSWI/SNF but not PBAF complexes[49]) and an in-house database of common contaminant using Mascot (Matrix Science, London, UK; version 2.6.2). Trypsin was selected as the enzyme, with one potential missed cleavage. Precursor ion tolerance was set to 10 ppm and fragment ion tolerance to 0.6 Da. Carbamidomethylation of cysteine was specified as fixed modification. Variable amino acid modification was oxidized methionine. Peptide-spectrum matches were validated using Percolator a target FDR of 0.01 and a Delta Cn of 0.5. For label-free quantification, features and chromatographic peaks were detected using the "Minora Feature Detector" Node with the default parameters. PSM and peptides were filtered with a false discovery rate (FDR) of 1%, and then grouped to proteins with again an FDR of 1% (strict) or 5% (relaxed) and using peptides with high confidence level. Low abundance resampling was used for missing value imputation. Both unique and razor peptides were used for quantitation and protein abundances are calculated as the average of the three most abundant distinct peptide group. The abundances were normalized on the "Total Peptide Amount" and then "Protein abundance based" option was selected for protein ratio calculation and associated *p*-values were calculated with an ANOVA test (individual proteins) and adjusted using Benjamini Hochberg correction.

### Statistical analysis
The statistical tests performed are disclosed in the corresponding Fig. legends. Statistical tests were performed using R.

### Reporting summary
Further information on research design is available in the Nature Portfolio Reporting Summary linked to this article.

## Data availability
The data supporting the findings of this study are available from the corresponding authors upon request. The next-generation sequencing data generated in this study have been deposited in the GEO server database under accession code GSE268206. The mass spectrometry proteomics data generated in this study have been deposited in the ProteomeXchange Consortium via the PRIDE partner repository database under accession codes PXD054392 and PXD061408. Additionally, we analyzed previously published SMARCA4 ChIP-seq from WT mESCs[49] and m6A-RIP-seq data from WT human ESCs[55]. Source data for the figures and Supplementary Figs. are provided as a Source Data file. Source data are provided with this paper.

## Code availability
All analyses were performed using previously published or developed tools, as indicated in Methods. Details on the bioinformatics pipeline and scripts used to run these published analysis tools can be found at https://github.com/NLykoskoufis/BraunLabPipeline.

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

## Acknowledgements

We would like to thank K. Majzoub, R. Pillai, J. Kirkland, G. Crabtree and M. Stoeber for valuable discussions, advice and protocols. We kindly acknowledge members of the RE.A.D.S platform for helping to establish several assays as well as members of High Performance Computing (HPC) team at the University of Geneva for their support. Many thanks to R. Murr for his generosity in sharing numerous lab reagents and equipment as well as the WT mES cell line. We thank P. Herrera for sharing the DT-A expression plasmid. We thank our colleagues and group members in the Department of Genetic Medicine and Development for critical comments on the project and manuscript, in particular G. Andrey and P. Wu. This work was supported by the Swiss National Science Foundation grant PCEFP3_194305 (awarded to S.M.G.B.) and the iGE3 PhD fellowship (awarded to H. Schwaemmle).

## Author contributions

H. Schwaemmle designed and conducted experiments, analyzed data, designed figures and wrote the manuscript. H. Soldati developed cell lines, recruitment assays and analyzed data. N.L. performed bioinformatics analysis of genomics datasets. M.D. performed next generation sequencing experiments. A.H. designed, conducted and performed bioinformatics analysis of the proteomics experiments. S.M.G.B. conceived the project, designed and conducted experiments, analyzed data and wrote the manuscript.

## Competing interests

The authors declare no competing interests.
