## [Transparent Peer Review file · Nature Communications]

CRISPR screen decodes SWI/SNF chromatin remodeling complex assembly

Corresponding Author: Professor Simon Braun

Version 1:

Reviewer comments:

Reviewer #1

(Remarks to the Author)

Schwaemmle et al performed a genome-wide loss-of-function CRISPR screen with epigenome editing in murine embryonic stem cells to identify regulators of SWI/SNF complex activity. This screen featured a Nkx2.9 promoter-driven live/dead reporter system that was combined with an inducible SWI/SNF recruitment tool to screen for novel regulators of SWI/SNF complex transcriptional activation at bivalent promoters. Several hits emerged from this screen and were validated by individual gene CRISPR knock-outs. Two of the strongest hits, Mlf2 and Rbm15, were chosen for further characterization. Using the dTAG-13 system paired with RNA- and ATAC-seq, the authors studied the newly-identified role of MLF2 in SWI/SNF regulation by comparing the effects of SMARCA4 loss and MLF2 loss on the mESC transcriptome and chromatin accessibility landscape. They find that MLF2 regulates the transcription of a subset of SWI/SNF target genes and that MLF2 also regulates accessibility at SWI/SNF-targeted developmental enhancers in a manner independent of direct chromatin binding. They also characterize the link between RBM15- and SWI/SNF-dependent gene expression using Rbm15 KO and RBM15 Δ RRM1 mutants paired with mRNA-seq, m6A-RIP-seq, and V5-SWI/SNF IP mass-spectrometry. They find that the m6A RNA methylation pathway regulates the stoichiometry of SWI/SNF complex subunits and that these altered SWI/SNF subunit protein levels are correlated with dysregulated m6A states in RBM15 KO cells. Data from the RBM15 Δ RRM1 mutant suggests that the RRM1 domain contributes to the selectivity of m6A writer complexes. Global analysis of m6A-RIP-seq data identifies widespread m6A dysregulation in RBM15 KO cells across many classes of chromatin regulators which implies that defective m6A deposition pathways could have strong effects on epigenome stability.

Overall, this study presents an innovative live/dead screening system with advantages over FACS-based readouts. The validation of screen hits demonstrates the effectiveness of this approach, and the use of synthetic chromatin regulator targeting tools could be modified to identify novel regulators of other chromatin regulator complexes. Regulation of SWI/SNF by MLF2 and RBM15 is an interesting finding, however the characterization of these regulatory links is mostly descriptive and correlative, especially in the case of MLF2. Additional experiments should be performed to strengthen the conclusions of this work, particularly at the protein level.

Specific Comments:

1. The evidence surrounding Mlf2's role in SWI/SNF regulation is purely correlative. Additional experiments should be performed to better test the authors' suggestion that Mlf2 chaperone complex activity is affecting SWI/SNF transcriptional regulation.
 - 1a. The authors should knockout MLF2 and show SWI/SNF protein levels by western blot (similar to Figure 6B.). If MLF2 regulates SWI/SNF complex stability and assembly, there should be differences at the protein level between MLF2 KO and WT mESCs.
 - 1b. The authors should perform co-immunoprecipitation of Mlf2 to determine if it interacts with SWI/SNF subunits at the protein level
 - 1c. The authors should perform SMARCA4 ChIP-seq in MLF2 KO cells to determine how the KO affects global SWI/SNF chromatin targeting. The correlation of changed MLF2 KO ATAC sites and public SMARCA4 ChIP-seq data is interesting, but would be strengthened by directly profiling SMARCA4 in the context of MLF2 KO.
2. The findings that RBM15 KO affects the m6A state of many other chromatin regulators is interesting but protein-level data would strengthen this aspect of the manuscript.
 - 2a. The authors should perform western blots on some of these chromatin regulators in RMB15 KO cells to provide evidence

for their suggestion that m6A RNA methylation is a general mechanism to control subunit dosage and stoichiometry of chromatin regulator complexes

Minor comments:

- "Spectrometry" is misspelled in figure 6D and 6E.

Reviewer #2

(Remarks to the Author)

In this study, Schwaemmle and colleagues performed a CRISPR screen for BAF complex regulators. They use an elegant reporter cell line to inducibly recruit SS18-containing BAF complex (possibly a mix of cBAF and ncBAF), transcriptionally activating a gene that leads to cellular death. They can then perform a CRISPR-KO screen where genes required for BAF function in transcriptional activation led to a loss of transcription of the reporter gene and cells now survive. The paper then focuses on two of their top hits, MLF2 and RBM15. MLF2 is a poorly characterized chaperone protein, and the authors describe its role in promoting BAF activity as a subset of BAF-dependent genes. RBM15, part of the m6A RNA methylation writer complex, controls m6A modifications on specific SWI/SNF mRNAs to regulate protein levels of these subunits. These two proteins provide entirely unique mechanisms of BAF complex regulation in a cell, a testament to the strength of the authors' unique CRISPR screen-based approach. The evidence for their molecular mechanisms is strong, but the mechanistic insights into MLF2 could be further improved with the experiments listed below. Overall, I am very enthusiastic about this study and its potential interest in the chromatin and epigenetic disease fields.

Major:

"To test if MLF2 is required for BAF recruitment to chromatin we performed MLF2-V5 ChIP-seq in mESCs using the V5-tag inserted with the FKBPv degron system"

This is not what an MLF ChIP is actually testing. The MLF ChIP tests whether it binds to chromatin, and a correlation with SMARCA4 binding sites can be made. However, it does not address causality (direct or indirect). To test this, the authors should perform a SMARCA4 ChIP in an MLF + d13 condition. If global losses of SMARCA4 binding to chromatin are expected, this experiment must be carefully controlled with a spike-in.

"Since MLF2 does not modulate BAF activity by binding chromatin"

How can you be sure the V5 epitope isn't just hidden in a ChIP for MLF2? I'd be more cautious about interpreting a lack of ChIP signal, meaning it doesn't bind chromatin. Orthogonal assays could support this theory. The authors should perform nuclear fractionation to test whether MLF2 is present in the nucleoplasmic fraction and not the chromatin fraction. See PMID: 35749589 for methods pertaining to other chromatin regulators, such as the BAF complex. Alternatively, or additionally, sequential salt extractions of chromatin could provide insight into MLF2 chromatin binding strength (see PMID: 28994797).

Minor:

"First, a module containing SMARCC1/2 dimers, SMARCD1/2/3, SMARCE1 and SMARCB1 is formed. Next, the ARID1A/B and DPF1/2/3 module is added to form the core of the BAF complex."

You are specifically describing cBAF assembly here. You should be specific and use cBAF instead of BAF when appropriate. I know this can be hard with SS18 recruitment, which could also be ncBAF, so it may not always be possible.

"It would therefore be interesting to isolate PBAF complexes from RBM15 KO cells to determine if misregulation of m6A levels also impacts PBAF complex assembly."

I agree, and while outside the scope of this study, I encourage the group to pursue this avenue of study.

Reviewer #3

(Remarks to the Author)

In this study, Schwaemmle et al. designed an innovative reporter system to identify regulators of the SWI/SNF chromatin remodelling complex using a CRISPR screen. They further characterised the roles of two distinct hits, MLF2 and RBM15. Overall, I find this study very interesting, with clear data presentation and seemingly robust results. I have raised several major and minor concerns for the authors and hope that my comments will help improve the clarity of the manuscript.

Major Concerns

1, Clarity on MLF2's Role. The model describing how MLF2 regulates SWI/SNF is not entirely clear. The authors state that "MLF2 does not regulate SWI/SNF targeting but rather modulates upstream processes such as SWI/SNF complex stability and assembly." If this is accurate, the amplitude of differential gene expression and ATAC accessibility should be similar, or only slightly weaker, in acute MLF2 depletion compared to acute SMARCA4 depletion. Additionally, these effects should be uniformly distributed across all SWI/SNF-targeted genes, rather than being concentrated in genes with high SMARCA4 occupancy. Could the authors clarify their results in this context?

2, m6A Regulation and SWI/SNF mRNAs. There seems to be a discrepancy regarding the m6A regulation on SWI/SNF mRNAs. The authors identified RBM15, VIRMA, and YTHDF1/2/3 as top hits and validated the effects using METTL3 inhibitors, exploring the potential role of RBM15 further. However, I have a few concerns: (i) Global m6A Reduction: While METTL3 inhibition likely causes a global reduction in m6A, why were METTL3 and METTL14 not identified as top hits alongside RBM15? The authors acknowledge that RBM15 targets the m6A writer complex to achieve site-specific methylation rather than acting as a global regulator. They also noted that m6A peaks are up- and down-regulated upon

RBM15 knockout. Although both RBM15 and METTL3 are involved in m6A regulation, they seem to play different roles or target different mRNAs within the SWI/SNF complex. Could the authors elaborate on this difference? (ii) RBM15B's Role: Given that RBM15B is also expressed in mESCs, why was RBM15B not identified in the CRISPR screen? Do the authors think that RBM15 and RBM15B have different mRNA target preferences? What will happen if both RBM15 and RBM15B are knocked out, regarding the regulation of SWI/SNF complex? (iii) Translation to Human Cells: The authors mention that m6A deposition can be influenced by gene architecture, such as exon/intron structure and the EJC complex. Since this study was conducted in mouse ES cells, how transferable are these insights to human cells? Could there be differences in the exon/intron architecture of SWI/SNF mRNAs between human and mouse that might affect these findings?

Minor Concerns

- 1, It's unclear why the authors chose these specific 9 genes. Not all of these genes are shown in Figure 1e. Could the authors clarify the criteria for selecting these genes and explain why some are not depicted in the figure.
- 2, There are several typos for gene names. Can the authors check the spelling for the following ones throughout the whole manuscript, Ythdf1, Ythdf2, Ythdf3, YTHDF1/2/3, METTL3 (page 10).
- 3, In Page 6, the authors use "50% of MLF2 target genes". How are MLF2 target genes defined? By ChIP-seq peaks or just differentially expressed genes upon knockout?

Reviewer #4

(Remarks to the Author)

General Overview:

In this study, Schwaemmle et al., performed a genome-wide screening to identify novel regulators of SWI/SNF complex and uncovered MLF2 and RBM15 as key modulators. They demonstrated that MLF2 is involved in regulation of chromatin accessibility of SWI/SNF complex, while RBM15 controls m6A modifications on specific SWI/SNF mRNAs to regulate protein level of these subunits. In general, this work provides valuable insights into new regulatory mechanisms of the SWI/SNF complex, but some of the major proteomics work should be carefully revisited.

To confirm SS18-V5-PYL1 expression recruits SWI/SNF complexes in mESCs, authors performed Co-immunoprecipitation (Co-IP) followed by mass spectrometry, presenting peptide counts from a single experiment (Supplementary Fig 1a.). As they appear to have enough ability to perform quantitative proteomics, I strongly recommend employing a robust quantitative approach, such as TMT or SILAC, with multiple replicates to ensure statistical rigor and enhance the overall quality of the study.

Regarding figure 6. The authors propose that misregulation on SWI/SNF mRNA in RBM 15 KO mESCs may lead to altered abundance of protein subunits, resulting in impaired SWI/SNF complex activity. To investigate this question regarding the measurement of protein abundance, authors performed IP-MS to measure abundance of SWI/SNF complex subunits. While the majority of translated subunits may form a complex, but it does not guarantee total pool of translated subunit proteins is reflected in the complex. Therefore, pulling down the complex still can provide a certain level of evidence to support authors idea, but it does not fully address the broader question of total protein abundance as a consequence of misregulated m6A levels.

Additionally, the authors performed western blots for several subunits normalizing the intensity of protein bands to Ponceau staining. However, Ponceau is not a highly sensitive quantification method, which could lead to inaccurate measurements and potentially affect the validity of their conclusions.

To ensure accurate protein quantification, I strongly recommend conducting an in-depth global proteomics experiment utilizing a more robust quantitative method, such as TMT or SILAC, rather than relying on LFQ, which may introduce missing values and compromise the reliability of the data.

Minor points

- Figure 6b, please show all the replicate western blots.
- Please revisit the method section to provide full details of the method, such as reagent used for reducing/alkylation and their concentration used, etc.
- Include accession information for deposited proteomics data

Version 2:

Reviewer comments:

Reviewer #1

(Remarks to the Author)

The authors have addressed my comments.

Reviewer #2

(Remarks to the Author)

Schwaemmle et al., have greatly improved their manuscript and addressed my concerns and questions in full. In particular, the SMARCA4 Cut&Run in the MLF-dTAG line significantly strengthened the robustness of their model. The sequential salt extractions were consistent with their model but once again strengthened it experimentally. Their rebuttal letter and manuscript changes were clear and easy to follow, making a secondary manuscript review straightforward. I found their

responses to the other reviewers strong. I fully support the publication of this study with two minor clarifications cited below:

Minor clarifications: Figure 4

Legend 4E: “centered at the TSS and ranked according to read density.” Are these centered on a TSS or the peak center? If at on the TSS, why exclude distal peaks off the TSS (like enhancers), which are expected to be a large portion of SMARCA4 peaks?

4F: The figure title (SMARCA4-Cut&Run) and the figure legend (ATAC-seq) do not match. It is a combination of experiments. Would “ATAC-seq signal at SMARCA4 sites” be a clearer title?

Reviewer #3

(Remarks to the Author)

I have carefully read the manuscript and the responses to my concerns. I appreciate the authors thorough responses to my previous concerns. I am satisfied with most of the explanations; however, I would like to seek further clarification regarding one specific point.

From the RNA-seq and MeRIP-seq data, it is evident that the m6A landscape is reshaped upon RBM15 knockout, rather than experiencing a global reduction. Specifically, some m6A sites are upregulated, while others are downregulated. The data presented suggest that this reshaping—such as the gain of m6A modification on the *Smarcc1* gene leading to increased protein levels—likely contributes to the observed phenotype, i.e. the assembly of incomplete complexes lacking the catalytic ATPase/ARP subunits.

I appreciate the authors' reference to the “potential” negative selection of METTL3 and METTL14 knockout cells, as presented in Chelmicki et al., Nature (2021). However, I would like to point out that the CRISPR-Cas screen and sgRNA characterization in that study were conducted after 5, 7, and 9 days selection. Additionally, while the authors employed STM2457 to inhibit METTL3's catalytic activity—an approach supported by multiple previous studies showing that STM2457 leads to global m6A reduction—it is important to consider that the observed survival of mESCs upon METTL3 inhibition may arise through mechanisms distinct from m6A landscape reshaping from RBM15 KO.

Interestingly, the study also identifies YTHDF1/2/3 and VIRMA as top candidates. As key cytosolic m6A readers, YTHDF1/2/3 facilitate m6A-containing RNA degradation via multiple pathways. I propose that the global m6A reduction resulting from METTL3 inhibition and the knockout of YTHDF1/2/3 may act through similar mechanisms, as both perturb global m6A regulation. However, as shown in Figure 5k, some genes exhibit increased m6A modification and RNA stability, suggesting that m6A enhances RNA stability in these cases. This raises the possibility that IGF2BP1/2/3, known m6A readers involved in RNA stabilization, may play a role.

I greatly appreciate the authors' efforts in characterizing the role of RBM15B, as this adds valuable insights not only to this study but also to the broader RNA modification field. While several m6A-related genes were identified in this study, they likely contribute to the observed phenotype through distinct mechanisms.

Reviewer #4

(Remarks to the Author)

I have thoroughly reviewed your rebuttal and the corresponding versions made to the manuscript.

I appreciate your effort in addressing all the comments and concerns raised during the review process.

The authors have addressed my concerns, and your well structured responses have significantly improved the clarity and rigor of the manuscript.

Point-by-point response to reviewer comments for manuscript NCOMMS-24-34565A-Z

Reviewer #1:

Overall, this study presents an innovative live/dead screening system with advantages over FACS-based readouts. The validation of screen hits demonstrates the effectiveness of this approach, and the use of synthetic chromatin regulator targeting tools could be modified to identify novel regulators of other chromatin regulator complexes. Regulation of SWI/SNF by MLF2 and RBM15 is an interesting finding, however the characterization of these regulatory links is mostly descriptive and correlative, especially in the case of MLF2. Additional experiments should be performed to strengthen the conclusions of this work, particularly at the protein level.

We thank the reviewer for their appreciation of our novel screening strategy and agree the recruitment tool is well suited to future studies of other chromatin regulators. We also thank the reviewer for the insightful comments that we have addressed to improve our manuscript, notably on the mechanism of MLF2-dependent regulation of SWI/SNF complexes.

1. The evidence surrounding Mlf2's role in SWI/SNF regulation is purely correlative. Additional experiments should be performed to better test the authors' suggestion that Mlf2 chaperone complex activity is affecting SWI/SNF transcriptional regulation.

1a. The authors should knockout MLF2 and show SWI/SNF protein levels by western blot (similar to Figure 6B.). If MLF2 regulates SWI/SNF complex stability and assembly, there should be differences at the protein level between MLF2 KO and WT mESCs.

1b. The authors should perform co-immunoprecipitation of Mlf2 to determine if it interacts with SWI/SNF subunits at the protein level.

1c. The authors should perform SMARCA4 ChIP-seq in MLF2 KO cells to determine how the KO affects global SWI/SNF chromatin targeting. The correlation of changed MLF2 KO ATAC sites and public SMARCA4 ChIP-seq data is interesting, but would be strengthened by directly profiling SMARCA4 in the context of MLF2 KO.

To address these points, we now include new data sets which directly support our proposed model that MLF2 acts as a chaperone to stabilize SWI/SNF complex assembly.

1a. In MLF2-dTAG mESCs treated with dTAG-13 for 8h, we measured SWI/SNF subunit protein levels via western blot. For the subunits tested (SMARCA4, SMARCC1, SMARCD1, SMARCE1 and ARID1A) we did not detect significant differences in dTAG-13 treated cells compared to controls. This data suggests that MLF2 acts as a chaperone to regulate SWI/SNF complex assembly rather than the abundance of individual subunits. We have included this new data in **Supplementary Fig. 3h**.

1b. We performed Proximity Ligation Assays (PLA) to determine if and where MLF2 and SMARCA4 interact in mESCs. This imaging technique allows *in situ* detection of endogenous protein interactions with high specificity and sensitivity. Using specific primary antibodies and oligonucleotide labelled secondary antibodies we were able to detect interactions between MLF2 and SMARCA4 in the nucleus of mESCs. As a control we depleted MLF2 using the dTAG system which caused a massive reduction in the PLA signal. We have included this new data in **Fig. 3h**.

1c. We performed SMARCA4 CUT&RUN experiments on MLF2-dTAG mESCs and found that loss of MLF2 leads to a global decrease in SWI/SNF binding to chromatin. To measure direct effects on SWI/SNF binding we performed the CUT&RUN experiments after 8h of MLF2 degradation, the same conditions that were used for the ATAC-seq experiments. To ensure accurate quantification of SWI/SNF binding across the three replicates we included spike-in controls in each sample. We have included this new data in **Fig. 4e-g** and **Supplementary Fig. 4d**.

Taken together these data provide direct evidence that the chaperone MLF2 interacts with SWI/SNF complexes to promote SWI/SNF complex assembly, which is required for proper binding to chromatin. This model is further supported by our previous data showing overlap between MLF2-dependent and SWI/SNF-dependent changes in chromatin accessibility and gene expression.

2. The findings that RBM15 KO affects the m6A state of many other chromatin regulators is interesting but protein-level data would strengthen this aspect of the manuscript.

2a. The authors should perform western blots on some of these chromatin regulators in RBM15 KO cells to provide evidence for their suggestion that m6A RNA methylation is a general mechanism to control subunit dosage and stoichiometry of chromatin regulator complexes.”

We agree that extending the role of RBM15 beyond the regulation of SWI/SNF mRNAs is an interesting addition to the manuscript. Thus, we measured changes in protein levels of several chromatin regulators which showed upregulation of m⁶A levels in the RBM15 KO mESCs. We found that the protein levels of the histone methyltransferase ASH2L and the DNA methyltransferase DNMT1 were also upregulated in RBM15 KO mESCs, whereas the levels of the histone methyltransferase EZH2 and the histone deacetylase HDAC1 were unchanged. These data suggest that, as for the SWI/SNF complex, the abundance of a subset of chromatin regulator subunits is regulated by m⁶A deposition by RBM15 and the m⁶A-writer complex. We have included this new data in **Supplementary Fig. 6g**.

Minor comments: “Spectrometry” is misspelled in figure 6D and 6E.

Thank you for pointing out these typos.

Reviewer #2:

In this study, Schwaemmle and colleagues performed a CRISPR screen for BAF complex regulators... These two proteins provide entirely unique mechanisms of BAF complex regulation in a cell, a testament to the strength of the authors' unique CRISPR screen-based approach. The evidence for their molecular mechanisms is strong, but the mechanistic insights into MLF2 could be further improved with the experiments listed below. Overall, I am very enthusiastic about this study and its potential interest in the chromatin and epigenetic disease fields.

We thank the reviewer for their enthusiasm for our study and for the experiments suggested to improve our understanding of the MLF2 mechanism. As mentioned for reviewer 1, we now include new datasets which directly support our proposed model that MLF2 acts as a chaperone to stabilize SWI/SNF complex assembly.

“To test if MLF2 is required for BAF recruitment to chromatin we performed MLF2-V5 ChIP-seq in mESCs using the V5-tag inserted with the FKBPv degen system.”

This is not what an MLF ChIP is actually testing. The MLF ChIP tests whether it binds to chromatin, and a correlation with SMARCA4 binding sites can be made. However, it does not address causality (direct or indirect). To test this, the authors should perform a SMARCA4 ChIP in an MLF + d13 condition. If global losses of SMARCA4 binding to chromatin are expected, this experiment must be carefully controlled with a spike-in.

To test for causality and a direct link between MLF2 function and SWI/SNF binding, we performed SMARCA4 CUT&RUN experiments on MLF2-dTAG mESCs and found that loss of MLF2 leads to a global decrease in SWI/SNF binding to chromatin. To measure direct effects on SWI/SNF binding we performed the CUT&RUN experiments after 8h of MLF2 degradation, the same conditions which were used for the ATAC-seq experiments. To ensure accurate quantification of SWI/SNF binding across three replicates we included spike-in controls in each sample. Furthermore, we performed positive (H3K4me3) and negative (IgG) controls to validate the CUT&RUN experiments. We have included this new data in **Fig. 4e-g** and **Supplementary Fig. 4d**.

“Since MLF2 does not modulate BAF activity by binding chromatin.”

How can you be sure the V5 epitope isn't just hidden in a ChIP for MLF2? I'd be more cautious about interpreting a lack of ChIP signal, meaning it doesn't bind chromatin. Orthogonal assays could support this theory. The authors should perform nuclear fractionation to test whether MLF2 is present in the nucleoplasmic fraction and not the chromatin fraction. See PMID: 35749589 for methods pertaining to other chromatin regulators, such as the BAF complex. Alternatively, or additionally, sequential salt extractions of chromatin could provide insight into MLF2 chromatin binding strength (see PMID: 28994797).

We agree that lack of ChIP-seq signal is not sufficient proof to conclude that a protein does not bind chromatin. We thus fractionated proteins from the cytoplasm and the nucleus of MLF2-FKBPv-V5 mESCs. Using the nuclear lysates, we then performed the sequential salt extraction assay described in PMID 28994797. We find that MLF2 is detected in the cytoplasmic and nuclear fractions but is not eluted from chromatin bound fractions at any of the salt concentrations (100-500nM NaCl). As a control we show SMARCA4 and CTCF are tightly bound to chromatin and can be sequentially eluted with buffers containing high salt concentrations. We have included this new data in **Supplementary Fig. 4f**.

Minor:

“First, a module containing SMARCC1/2 dimers, SMARCD1/2/3, SMARCE1 and SMARCB1 is formed. Next, the ARID1A/B and DPF1/2/3 module is added to form the core of the BAF complex.” You are specifically describing cBAF assembly here. You should be specific and use cBAF instead of BAF when appropriate. I know this can be hard with SS18 recruitment, which could also be ncBAF, so it may not always be possible.

We agree that only referring to SWI/SNF complexes may lead to confusion and have therefore included a description of the cBAF, ncBAF and PBAF sub-complexes on **page 4**. In addition, we now include new proteomics data which shows that by using the SS18 subunit to recruit SWI/SNF, we efficiently pull down cBAF and ncBAF sub-complexes but not PBAF. We have included this new data in **Supplementary Fig. 1a**.

“It would therefore be interesting to isolate PBAF complexes from RBM15 KO cells to determine if misregulation of m6A levels also impacts PBAF complex assembly.” I agree, and while outside the scope of this study, I encourage the group to pursue this avenue of study.

We agree and are planning future studies where we will use PHF10-tagged subunits to specifically recruit PBAF and perform another KO CRISPR screen.

Reviewer #3:

In this study, Schwaemmle et al. designed an innovative reporter system to identify regulators of the SWI/SNF chromatin remodelling complex using a CRISPR screen. They further characterised the roles of two distinct hits, MLF2 and RBM15. Overall, I find this study very interesting, with clear data presentation and seemingly robust results. I have raised several major and minor concerns for the authors and hope that my comments will help improve the clarity of the manuscript.

Many thanks to the reviewer for their interesting comments and suggestions for revision experiments which have strengthened our manuscript.

1, Clarity on MLF2's Role. The model describing how MLF2 regulates SWI/SNF is not entirely clear. The authors state that “MLF2 does not regulate SWI/SNF targeting but rather modulates upstream processes such as SWI/SNF complex stability and assembly.” If this is accurate, the amplitude of differential gene expression and ATAC accessibility should be similar, or only slightly weaker, in acute MLF2 depletion compared to acute SMARCA4 depletion. Additionally, these effects should be uniformly distributed across all SWI/SNF-targeted genes, rather than being concentrated in genes with high SMARCA4 occupancy. Could the authors clarify their results in this context?

As described in our responses to Reviewers 1 and 2, we have now performed new experiments to support our model and clarify our initial results. First, we show via a proximity ligation assay that MLF2 and SMARCA4 interact in the nucleus of mESCs. Next, using salt fractionation assays we show that MLF2 does not bind chromatin. These data suggest that MLF2 likely acts as a chaperone to promote SWI/SNF complex assembly, upstream of SWI/SNF's role on chromatin. Next, we performed SMARCA4 CUT&RUN in MLF2-dTAG cells and found that SWI/SNF chromatin binding is reduced genome-wide after rapid MLF2-depletion. Thus, the effects are indeed uniformly distributed across all SWI/SNF binding sites. However, as mentioned by the reviewer, this does not lead to equivalent genome-wide changes in chromatin accessibility and gene expression. Because MLF2 is not essential for SWI/SNF assembly but acts rather as a modulator of complex assembly, its depletion does not lead

to a complete loss in SWI/SNF activity as in the SMARCA4-depleted cells. Therefore, we propose that only the subset of loci/genes that depend almost exclusively on high levels of SWI/SNF activity are impacted by MLF2 depletion. Indeed, we show that the sites which show changes in chromatin accessibility upon MLF2 loss are characterized by high levels of SMARCA4 occupancy in mESCs. These results also highlight the activity of the many different and often redundant chromatin regulators that control chromatin accessibility in addition to the SWI/SNF complex. We have included these new datasets in **Fig. 3h**, **Fig. 4e-g** and **Supplementary Fig. 4d,f**.

2, m6A Regulation and SWI/SNF mRNAs. There seems to be a discrepancy regarding the m6A regulation on SWI/SNF mRNAs. The authors identified RBM15, VIRMA, and YTHDF1/2/3 as top hits and validated the effects using METTL3 inhibitors, exploring the potential role of RBM15 further. However, I have a few concerns:

(i) Global m6A Reduction: While METTL3 inhibition likely causes a global reduction in m6A, why were METTL3 and METTL14 not identified as top hits alongside RBM15? The authors acknowledge that RBM15 targets the m6A writer complex to achieve site-specific methylation rather than acting as a global regulator. They also noted that m6A peaks are up- and down-regulated upon RBM15 knockout. Although both RBM15 and METTL3 are involved in m6A regulation, they seem to play different roles or target different mRNAs within the SWI/SNF complex. Could the authors elaborate on this difference?

We agree with the statement that RBM15 and METTL3/14 play different roles in the m⁶A complex. The METTL3/14 subunits are the catalytic subunits of the complex whereas RBM15 likely regulates site specificity via its three RNA recognition motif (RRM) domains. Previous studies have shown that *Mettl3* and *Mettl14* KO mESCs display differentiation and lethality effects when grown for several days in the same media conditions as those used in our study (Chelmicki et al. Nature 2021, Extended Data Fig. 3f). It is therefore likely that we do not detect these genes enriched in our CRISPR screen as the pool of KO mESCs is first selected for 7 days before performing the recruitment assay to induce DT expression. However, to overcome this issue, we performed acute inhibition of the m⁶A-complex with the chemical inhibitor STM2457, and observed a significant increase in cell viability compared to controls after ABA treatment (**Fig. 5b**). Interestingly, in our genetic screen, we detect an enrichment of the *Ythdf1*, *Ythdf2* and *Ythdf3* genes, which are involved in downstream regulation of mRNA stability and translation rates.

(ii) RBM15B's Role: Given that RBM15B is also expressed in mESCs, why was RBM15B not identified in the CRISPR screen? Do the authors think that RBM15 and RBM15B have different mRNA target preferences? What will happen if both RBM15 and RBM15B are knocked out, regarding the regulation of SWI/SNF complex?

As mentioned by the reviewer, both *Rbm15* and *Rbm15b* are expressed in mESCs (**Supplementary Fig. 6b**). Therefore, to confirm that RBM15B was not missed in our CRISPR screen we generated *Rbm15b* KO mESCs and performed the SWI/SNF recruitment assay. We found that RBM15B loss did not rescue cell viability, suggesting that RBM15B does not regulate SWI/SNF activity. We also performed western blot analysis of SWI/SNF subunits in *Rbm15b* KO mESCs, and in contrast to *Rbm15* KO cells, we did not detect any differences in SWI/SNF subunits levels. These data suggest that RBM15 and RBM15B have different roles in mESCs, most likely due to their unique sets of RNA recognition motifs. We have included this new data in **Supplementary Fig. 6d-f**.

(iii) Translation to Human Cells: The authors mention that m6A deposition can be influenced by gene architecture, such as exon/intron structure and the EJC complex. Since this study was conducted in mouse ES cells, how transferable are these insights to human cells? Could there be differences in the exon/intron architecture of SWI/SNF mRNAs between human and mouse that might affect these findings?

We agree that as SWI/SNF misregulation is a major driver of human disease, it is interesting to discuss the translation of our mouse data to human cells. The point raised by the reviewer was addressed in a recent study where the authors looked at how exon-junction architecture regulates mRNA abundance through m⁶A methylation across different organisms (*He et al., Science 2023*, Supplementary Figure 23). Their analysis suggests that through evolution, the correlation between exon-intron architecture and mRNA stability holds for fish, mice and humans but not for flies and worms. To focus specifically on m⁶A regulation of SWI/SNF mRNAs between mice and humans, we analyzed published m⁶A-RIP-

seq data from human ES cells and compared it to our mouse ES cell data. We found that as in mice, many human SWI/SNF mRNAs were m⁶A-modified in ES cells, with numerous SWI/SNF subunits displaying similar patterns of m⁶A modifications in both species. We have included this new data in **Supplementary Fig. 7**.

Minor Concerns

1, It's unclear why the authors chose these specific 9 genes. Not all of these genes are shown in Figure 1e. Could the authors clarify the criteria for selecting these genes and explain why some are not depicted in the figure.

Indeed, the 9 genes selected were not only chosen based on their rank in the Enrichment score obtained from the MAGeCK analysis (**Fig. 1e**). In addition to high fold change and low p-value criteria, we selected genes based on these criteria: (1) no clear link to SWI/SNF activity (*Mif2/Rbm15/Gna13*), (2) known regulators of bivalent gene expression (*Setd1b/Kdm6a/Tet1*), (3) known transcription factor (*Hopx, Tfap2c*), and (4) not previously identified in genetic screens using DT-mediated cell death as a readout (except *Dph2*). For a detailed overview of the fold enrichment measured for each of the six sgRNAs from both library A&B for the selected hits, please refer to **Supplementary Fig. 1e**.

2, There are several typos for gene names. Can the authors check the spelling for the following ones throughout the whole manuscript, Ythdf1, Ythdf2, Ythdf3, YTHDF1/2/3, METTL3 (page 10).

Thank you for pointing out these typos.

3, In Page 6, the authors use "50% of MLF2 target genes". How are MLF2 target genes are defined? By ChIP-seq peaks or just differentially expressed genes upon knockout?"

They are defined by differential expression upon knockout as MLF2 does not bind chromatin.

Reviewer #4:

In this study, Schwaemmle et al., performed a genome-wide screening to identify novel regulators of SWI/SNF complex and uncovered MLF2 and RBM15 as key modulators. They demonstrated that MLF2 is involved in regulation of chromatin accessibility of SWI/SNF complex, while RBM15 controls m6A modifications on specific SWI/SNF mRNAs to regulate protein level of these subunits. In general, this work provides valuable insights into new regulatory mechanisms of the SWI/SNF complex, but some of the major proteomics work should be carefully revisited.

We thank the reviewer for their comments on the valuable insights provided by our study. To address their concerns related to the proteomics data we include new datasets that confirm our initial results while improving the statistical rigor of the study. In brief, we increased the number of replicates in our immunoprecipitation mass spectrometry (IP-MS) measurements, and used a more robust and quantitative method for analysis of total protein abundance in control and RBM15 KO mESCs.

To confirm SS18-V5-PYL1 expression recruits SWI/SNF complexes in mESCs, authors performed Co-immunoprecipitation (Co-IP) followed by mass spectrometry, presenting peptide counts from a single experiment (Supplementary Fig 1a.). As they appear to have enough ability to perform quantitative proteomics, I strongly recommend employing a robust quantitative approach, such as TMT or SILAC, with multiple replicates to ensure statistical rigor and enhance the overall quality of the study.

We agree that the IP-MS data presented in **Supplementary Fig. 1a** did not meet the statistical rigor in line with the rest of the data presented in this study. To correct this, we performed the IP-MS analysis in triplicates and confirmed the enrichment of all SWI/SNF subunits in the immunoprecipitation assays. Using the MS1-based label-free quantitative (LFQ) method, we show that in cells expressing V5-SS18 we can efficiently IP the entire SWI/SNF complex using anti-V5 magnetic beads. In particular, we pulled down cBAF and ncBAF subcomplexes which contain SS18 but not the pBAF subcomplex which lacks SS18. In addition, we performed western blot analyses of several subunits to confirm that SS18-V5 co-immunoprecipitates with the SMARCC1, SMARCA4, SMARCD1 and SMARCE1 subunits. In control cells not expressing V5-SS18, we do not pull down any SWI/SNF subunits using the anti V5-beads,

highlighting the specificity of the immunoprecipitation protocol. Together with the V5 ChIP-qPCR, SMARCC1 ChIP-qPCR and ATAC-qPCR data shown in **Fig. 1b** and **Supplementary Fig. 1c**, these results show that our reporter cell line efficiently recruits intact and functional SWI/SNF complexes to the *Nkx2-9* locus upon ABA-treatment. We have included the new proteomics and western blot data in **Supplementary Fig. 1a,b**.

Regarding figure 6. The authors propose that misregulation on SWI/SNF mRNA in RBM 15 KO mESCs may lead to altered abundance of protein subunits, resulting in impaired SWI/SNF complex activity. To investigate this question regarding the measurement of protein abundance, authors performed IP-MS to measure abundance of SWI/SNF complex subunits. While the majority of translated subunits may form a complex, but it does not guarantee total pool of translated subunit proteins is reflected in the complex. Therefore, pulling down the complex still can provide a certain level of evidence to support authors idea, but it does not fully address the broader question of total protein abundance as a consequence of misregulated m6A levels. Additionally, the authors performed western blots for several subunits normalizing the intensity of protein bands to Ponceau staining. However, Ponceau is not a highly sensitive quantification method, which could lead to inaccurate measurements and potentially affect the validity of their conclusions. To ensure accurate protein quantification, I strongly recommend conducting an in-depth global proteomics experiment utilizing a more robust quantitative method, such as TMT or SILAC, rather than relying on LFQ, which may introduce missing values and compromise the reliability of the data.

As mentioned by the reviewer, the IP-MS data presented in our study reveals that in RBM15 KO mESCs, the abundance of SWI/SNF subunits is altered in purified complexes. The increase in core subunits leads to incomplete complex assembly, revealing the RBM15-dependent mechanism which is responsible for the impaired SWI/SNF activity initially identified in the genetic screen. While this is the main goal of the proteomics analysis, it is true that it does not address the total abundance of subunits within and outside of SWI/SNF complexes. To address this point, we decided to focus on an improved method for total protein quantification while still relying on western blots imaged with the Licor Odyssey system, which remains a highly sensitive approach for measuring differences in the abundance of specific proteins. Indeed, instead of the Ponceau stain, we now use a fluorescence based total protein stain for western blot normalization. With this new sensitive method, we can accurately measure total protein abundance on the Licor Odyssey imaging system before immunostaining of the membranes. As the RBM15 KO shows major changes in gene expression, we chose not to rely on housekeeping proteins such as GAPDH for normalization but rather a more robust total protein normalization. Importantly, before loading cell lysates on the SDS-PAGE gels we also perform high precision BCA protein assays to ensure equal amounts of protein are loaded in each well. As shown in the uncropped western blots below, the data shows that the total abundance of core SWI/SNF subunits SMARCC1 and ARID1A is increased in RBM15 KO mESCs. These data confirm our previous results obtained with the Ponceau staining method for total protein quantification, with the only difference being ACTINB which is no longer significantly reduced. We have included this new data in **Fig. 6b,c** and **Supplementary Fig. 6c**.

Minor points

- Figure 6b, please show all the replicate western blots.

Please see below the uncropped replicate western blots. For comparison, we included the results from both the initial Ponceau staining method and the new fluorescent Revert 520 total protein staining method from LICORbio.

- Please revisit the method section to provide full details of the method, such as reagent used for reducing/alkylation and their concentration used, etc.

For the proteomics methods, we used the iST kit from Preomics to process the IP samples and unfortunately it does not include any information on the reduction/alkylation reagents or their concentration. If required we can contact the company to request this information.

- Include accession information for deposited proteomics data.

The mass spectrometry proteomics data have been deposited to the ProteomeXchange Consortium via the PRIDE partner repository with the dataset identifier **PXD054392**.

520 total protein stain

800, rabbit

700, mouse

Ponceau S total protein stain

800, rabbit

700, mouse

Point-by-point response to reviewer comments for manuscript NCOMMS-24-34565B

Reviewer #2:

Thank you for pointing out these errors that we have corrected. The figure legends now read:

Figure 4.

e. SMARCA4 Cut&Run signal in MLF2 dTAG cells with 8h treatment of DMSO or dTAG-13, centered at the peak center and ranked according to read intensity.

f. SMARCA4 Cut&Run signal at changed and unchanged ATAC peaks in MLF2-dTAG mESCs after 8h of dTAG-13 treatment.

Reviewer #3:

Thank you for your supportive comments on our revised manuscript, as well as your insights into m6A regulation.

We agree that the observed gains and increases in m6A distribution is likely specific to RBM15 KO and therefore may represent a distinct mechanism to METLL3 inhibition or KO of YTHDF1/2/3, both of which have been shown to lead to global reductions in m6A. To highlight this point, we have now included the following statement in the discussion:

Page 11: "We also identified the *Ythdf1*, *Ythdf2* and *Ythdf3* m⁶A-readers in the genome-wide screen for SWI/SNF regulators. These proteins have previously been shown to regulate both the stability and translation efficiency of m⁶A-methylated mRNAs. Consequently, the absence of these m⁶A-readers may also alter the abundance of specific SWI/SNF subunit mRNAs and impair complex assembly, or the observed phenotype may act through a distinct mechanism as YTHDF1/2/3 readers have been shown to impact global m⁶A regulation."

Also, as mentioned by the reviewer, our data suggests that m6A enhances RNA stability in certain cases, and raises the possibility that IGF2BP1/2/3 readers may also be involved. We therefore included this family of readers in the results section of our manuscript:

Page 7: "After methylation, m⁶A reader proteins like the YTHDF1/2/3 and IGF2BP1/2/3 families regulate the stability and translation efficiency of m⁶A-modified mRNAs."